# GRAPH NEURAL NETWORKS GONE HOGWILD

**Olga Solodova, Nick Richardson, Deniz Oktay, Ryan P. Adams**
Department of Computer Science
Princeton University
Princeton, NJ, USA
`{solodova, njkrichardson, doktay, rpa}@princeton.edu`

## ABSTRACT

Graph neural networks (GNNs) appear to be powerful tools to learn state representations for agents in distributed, decentralized multi-agent systems, but generate catastrophically incorrect predictions when nodes update asynchronously during inference. This failure under asynchrony effectively excludes these architectures from many potential applications where synchrony is difficult or impossible to enforce, e.g., robotic swarms or sensor networks. In this work we identify "implicitly-defined" GNNs as a class of architectures which is provably robust to asynchronous "hogwild" inference, adapting convergence guarantees from work in asynchronous and distributed optimization. We then propose a novel implicitly-defined GNN architecture, which we call an *energy GNN*. We show that this architecture outperforms other GNNs from this class on a variety of synthetic tasks inspired by multi-agent systems.

## 1    INTRODUCTION

Coordination and control of distributed and decentralized multi-agent systems is a major project spanning engineering and computational science. Success in this program has implications across a broad spectrum of applications; autonomous vehicle navigation, energy/resource management and distribution in smart grids, and robotic swarms (for exploration, search and rescue, environmental monitoring, and construction), to name a few. Agents in these systems must take action based on direct observation and communication with the collective. At the level of any individual, a controller takes as input the agent's 'state representation', which ideally unifies direct measurements and peer-derived messages in a coherent manner.

In the distributed/decentralized regime, the communication constraints associated with the system can be encoded as a graph in which each node is associated with an agent, and each edge with an agent-to-agent communication link. Adopting the graph view of the system, graph neural networks (in particular, message-passing variants (Gilmer et al., 2017; Hamilton, 2020)) have been widely studied as methods for deriving flexible, data-dependent, and parametric state representations. GNN-based state representations have been explored in a variety of applications; for example, flock formation, target tracking, path planning, goal assignment, and channel allocation in wireless networks (Jiang and Lu, 2018; Nakashima et al., 2019; Li et al., 2019; Khan et al., 2019; Jiang et al., 2020; Gama et al., 2020; Blumenkamp et al., 2021; Grattarola et al., 2021; Zhou et al., 2021; Gosrich et al., 2022; Jiang et al., 2023; Goarin and Loianno, 2024). The motivation to use GNNs can be traced to three core features. First, a GNN is a parametric family of functions for deriving a state representation, enabling domain-specificity by choosing from this family and optimizing the state representation for a given task end-to-end. Second, message passing within a given GNN layer satisfies the constraint that agents must derive state representations using only information obtained through local communication with neighbors. Third, GNNs with multiple layers allows state representations to depend on information outside of a given agent's local neighborhood.

All that said, there is a major conundrum in applying GNNs to these problems. On the one hand, asynchronous execution and unreliable communication are ubiquitous in real-world decentralized/distributed multi-agent systems. On the other hand, these features render conventional GNNs inoperable in the multi-layer regime, the very setting in which GNNs offer a putative advantage through longer range communication/coordination. This is because GNN architectures implicitly assume that a synchronization barrier is enforced across all nodes between layers of message passing. If nodes update asynchronously or if messages can be delayed or lost, embeddings from neighbors do not necessarily correspond with the intended layer in the GNN (from the perspective of the receiving node). The effective architecture (and therefore output) of the GNN diverges catastrophically from that used in training. Figure 1 illustrates the issue on a toy problem.

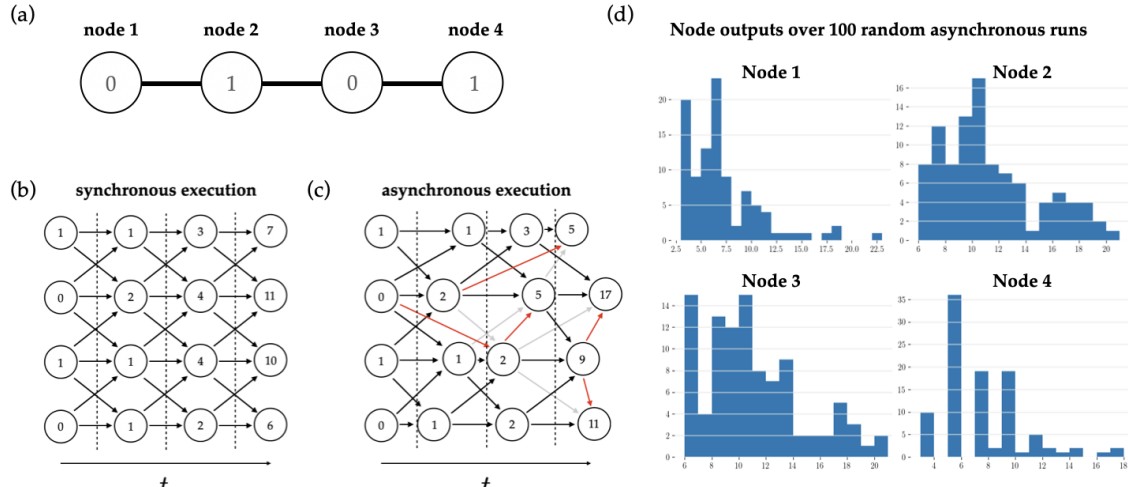

Figure 1: Modifications to the computation graph of an $L$-layer message-passing GNN resulting from asynchronous, distributed, per-node inference with communication delays. **(a)** An undirected linear graph with binary features written in the node body. **(b)** Synchronous per-layer execution of a 3-layer GNN on the graph from (a); all nodes update at the same time, using neighbor information from the previous layer. Arrows represent network weights. For demonstration, weights are all equal to 1, and the value of a node embedding at the next layer is the sum of incoming values. **(c)** Asynchronous inference of the GNN from (b). Nodes update at random times and can use neighbor information corresponding to the incorrect layer, which introduces modifications to the computation graph. Gray arrows correspond to connections that were removed from the original computation graph, and red arrows are unintended connections resulting from asynchrony and/or message delays. **(d)** To demonstrate the effect of asynchrony, we show the output of the GNN varies significantly over asynchronous runs with different node update orderings.

Unfortunately, enforcing synchronization in a distributed and decentralized system is costly (e.g. throughput is limited by the slowest node) and is particularly difficult to enforce in systems where nodes can join or leave the network, or are prone to failure. Existing work on using GNNs for computing agent state embeddings either use single layer GNNs (where synchronization is unnecessary), or ignore the constraint of asynchronous execution altogether, limiting the prospect of fielding the method. If it were possible to perform "hogwild" inference in multi-layer GNNs (nodding to the asynchronous optimization work of Niu et al. (2011)), GNNs would be a significantly more powerful tool in computing state embeddings for decentralized multi-agent systems.

In this work we present a unified framework for GNNs which can execute inference asynchronously, adapting classical results from the distributed optimization literature. Our analysis illuminates the boundary between GNNs which are/are not amenable to asynchronous inference, and we articulate sufficient conditions for a given architecture's robustness to asynchrony. We call the class of GNNs which are provably robust to asynchronous inference 'implicitly-defined'; in an implicitly-defined GNN, node embeddings correspond with a solution to an optimization problem (Gu et al., 2020; Liu et al., 2021; Yang et al., 2021; Zhu et al., 2021; Ma et al., 2021; Scarselli et al., 2009). This is in contrast to GNNs in which node embeddings are computed as the output of a statically defined feed-forward computation, such as graph attention networks (GAT, Veličković et al. (2018)) and graph convolutional networks (GCN, Kipf and Welling (2017)); we refer to these architectures as 'explicitly-defined'.

In addition to an analysis of GNNs under asynchronous execution, we contribute a novel asynchronous-capable (implicitly-defined) architecture, which we call energy GNN. Our architecture exposes a rich parameterization of optimization problems using input-convex neural networks operating over nodal neighborhoods. We show that energy GNNs outperform other implicitly-defined GNNs on a variety of synthetic tasks motivated by problems which are relevant to multi-agent systems, where synchronous inference may be undesirable. We also achieve competitive performance on tasks with benchmark graph datasets, certifying the merit of our approach even as a stand-alone GNN architecture.

## 2 PRELIMINARIES

### 2.1 PARTIALLY ASYCHRONOUS ALGORITHMS

Computational models for asynchronous algorithms vary depending on the constraints imposed on the sequencing or frequency of computation or communication. We consider *partial asynchronism* as defined by Bertsekas and Tsitsiklis (1989), which we summarize here.

Consider a collection of $n$ nodes carrying out a distributed computation. Each node has hidden state denoted $\boldsymbol{h}_i \in \mathbb{R}^k$, which corresponds to one "block" (row) of the aggregate state $\boldsymbol{H} := (\boldsymbol{h}_1, \ldots, \boldsymbol{h}_n)^T \in \mathbb{R}^{n \times k}$. The algorithm being executed consists of iterative node updates, where each node $i = 1, 2, \ldots, n$ iteratively updates its state according to $\boldsymbol{h}_i := f^i(\boldsymbol{H})$ using some node-specific update function $f^i : \mathbb{R}^{n \times k} \to \mathbb{R}^k$.

We are given a set $T^i \subseteq \{0, 1, 2, \ldots\}$ of times at which each node $i$ is updated; this accounts for asynchrony in the execution of the algorithm. Additionally, for each $t \in T^i$ we are given variables $0 \leq \tau_j^i(t) \leq t$ which represent the time associated with node $i$'s view of node $j$ at time $t$. If communication is instantaneous and reliable, then $\tau_j^i(t)$ is always equal to $t$, as node $i$'s view of all other nodes is the actual value those nodes hold at time $t$. Staleness (i.e. $\tau_j^i(t) < t$) results from delays or losses in communication between nodes. For instance, suppose $T^3 = \{0, 4, 7\}$, $T^1 = \{2, 3, 5\}$, and $\tau_1^3(4) = 2$. This means that when node 3 executes its second update (at $t = 4$), it uses the stale value of node 1 that was computed at $t = 2$ rather than its actual value at $t = 4$ (due to loss or delay of the message containing the value of node 1 which was computed at $t = 3$).

We can now describe the asynchronous algorithm using the following node update equations:

$$\boldsymbol{h}_i(t+1) = \boldsymbol{h}_i(t) \ \text{ if } t \notin T^i, \tag{1}$$

$$\boldsymbol{h}_i(t+1) = f^i(\boldsymbol{h}_1(\tau_1^i(t)), \ldots, \boldsymbol{h}_n(\tau_n^i(t))) \ \text{ if } t \in T^i, \tag{2}$$

for $t \geq 0$. Partial asynchronism corresponds to the following assumptions:

**Assumption 2.1 (Partial Asynchronism)**
*(Bertsekas and Tsitsiklis (1989) Assumption 5.3) There exists an integer $B > 0$ such that:*

(a) *(Bounded time to next update) For every node $i$ and for every $t \geq 0$, at least one of the elements of the set $\{t, t+1, \ldots, t + B - 1\}$ belongs to $T^i$.*

(b) *(Bounded staleness) There holds $t - B < \tau_j^i(t) \leq t$, for all $i, j$, and all $t \geq 0$ belonging to $T^i$.*

(c) *There holds $\tau_i^i(t) = t$ for all $i = 1, 2, \ldots, n$ and $t \in T^i$.*

Informally, (a) states that each node updates at least once every $B$ time units, (b) states that information from other nodes can be stale by at most $B$ time units, and (c) states that node $i$ maintains the current value of $\boldsymbol{h}_i$.

### 2.2 GRAPH NEURAL NETWORKS

Consider a directed graph with a collection of $n$ vertices $V = \{1, ..., n\}$, and edges $E \subseteq V \times V$. The connectivity of the graph is contained in its adjacency matrix $\boldsymbol{A} \in \{0, 1\}^{n \times n}$, where $\boldsymbol{A}_{i,j} = 1$ if there is an edge from node $i$ to node $j$, and 0 otherwise. The graph may also have associated node and edge features $\boldsymbol{X} \in \mathbb{R}^{n \times p}$ and $\boldsymbol{E} \in \mathbb{R}^{|E| \times r}$, so we use $\mathcal{G} = (\boldsymbol{A}, \boldsymbol{X}, \boldsymbol{E})$ to denote the graph. We use $\mathcal{N}(i)$ to denote the set of neighbors of node $i$.

In a GNN, each node $i$ is associated with a $k$-vector embedding $\boldsymbol{h}_i$ which is updated through iterations (or layers) of message passing. Each node constructs a "message" $\boldsymbol{m}_i$ by aggregating information from nodes $j \in \mathcal{N}(i)$ in its local neighborhood, then uses this aggregate message to update its embedding. The node update equation for all nodes $i$ and layers $\ell \in \{0, ..., L - 1\}$ is as follows:

$$\boldsymbol{h}_i^{\ell+1} = f_\theta^\ell\left(\boldsymbol{h}_i^\ell, \boldsymbol{x}_i, \{\boldsymbol{h}_j^\ell, \boldsymbol{x}_j, \boldsymbol{e}_{ij} | j \in \mathcal{N}(i)\}\right) = u_\theta^\ell\left(\boldsymbol{h}_i^\ell, \boldsymbol{x}_i, \bigoplus_{j \in \mathcal{N}(i)} m_\theta^\ell(\boldsymbol{h}_i^\ell, \boldsymbol{h}_j^\ell, \boldsymbol{x}_i, \boldsymbol{x}_j, \boldsymbol{e}_{ij})\right) = u_\theta^\ell\left(\boldsymbol{h}_i^\ell, \boldsymbol{x}_i, \boldsymbol{m}_i^\ell\right),$$

$$\tag{3}$$

where $\bigoplus$ is an aggregation function, and $u_\theta$ and $m_\theta$ are, e.g., neural networks. We use $\boldsymbol{H} := (\boldsymbol{h}_1, \ldots, \boldsymbol{h}_n)^T \in \mathbb{R}^{n \times k}$ to denote the aggregate embedding of nodes in the graph. $\boldsymbol{H}^0$ is typically initialized as the node features $\boldsymbol{X}$. For convenience, we omit the layer index $\ell$ from $f_\theta^\ell$ in contexts where there is only one parameterized layer, or when referring generally to a message passing layer. In the final layer, a readout function $o_\phi : \mathbb{R}^k \to \mathbb{R}^J$ is applied to each embedding, which results in node predictions $\hat{\boldsymbol{Y}} = (o_\phi(\boldsymbol{h}_1^L), ..., o_\phi(\boldsymbol{h}_n^L))^T \in \mathbb{R}^{n \times J}$.

# 3 EXPLICITLY-DEFINED VS. IMPLICITLY-DEFINED GNNS

In a conventional feed-forward message passing architecture that consists of $L$ layers, it is assumed that all update functions for a given layer are applied in a synchronized manner. That is, each node embedding is derived from the previous layer embeddings. In distributed systems synchrony is either impossible or comes with substantial overhead. If nodes update asynchronously according to Equations (1) and (2), GNN architectures that assume synchronicity will fail catastrophically and nondeterministically because the architecture at inference time is different than it was during training. This effect is illustrated in Figure 1.

This failure motivates us to distinguish between two types of GNN architectures: *explicitly-defined* GNNs which specify a specific feed-forward layer-wise computation, and *implicitly-defined* GNNs in which the layer-wise message passing updates correspond to iterations toward a fixed point. We further subdivide implicitly-defined GNNs into two types: *fixed-point* GNNs and the important special case of *optimization-based* GNNs. Explicitly-defined GNNs are susceptible to failure under asynchrony, while implicitly-defined GNNs are robust (as we discuss in Section 4).

## 3.1 FIXED-POINT GNNS

Fixed-point GNNs obtain node embeddings as the fixed point of a contractive message passing function. Using $\boldsymbol{h} \in \mathbb{R}^{nk}$ to denote the unrolled embeddings $\boldsymbol{H}$ and taking $F_\theta : \mathcal{G} \times \mathbb{R}^{nk} \to \mathbb{R}^{nk}$ to be the aggregate update of the hidden state from applying $f_\theta$ to the neighborhood subgraph of each node $i = 1, ...n$ in a graph $\mathcal{G}$, a fixed-point GNN iterates $F_\theta$ until numerical convergence of node embeddings, i.e., $F_\theta(\boldsymbol{h}) \approx \boldsymbol{h}$.

Convergence to a unique fixed point is guaranteed provided the update function $F_\theta$ is a contraction map with respect to the embeddings, i.e. $||F_\theta(\boldsymbol{h}) - F_\theta(\boldsymbol{h}')|| \leq \mu ||\boldsymbol{h} - \boldsymbol{h}'||$ holds for $0 < \mu < 1$ and $\forall \boldsymbol{h}, \boldsymbol{h}' \in \mathbb{R}^{nk}$. Since it is difficult to design non-trivial parameterizations of $F_\theta$ which are contractive by construction, existing fixed-point GNN architectures (e.g. IGNN (Gu et al., 2020), EIGNN (Liu et al., 2021), and APPNP (Gasteiger et al., 2019)) are limited in diversity. These architectures use linear transformations of the node embeddings in computing messages $\boldsymbol{m}_i$, where the parameters can easily be constrained such that $F_\theta$ is contractive. Gu et al. (2020) apply a component-wise non-expansive nonlinearity to compute updated node embeddings from these messages, while Liu et al. (2021); Gasteiger et al. (2019) directly take $\boldsymbol{m}_i$ as the node embeddings for the next iteration.

The original 'nonlinear GNN' proposed by Scarselli et al. (2009), in which $f_\theta$ contains general multi-layer neural networks operating on the input data, represents another approach to parameterizing a fixed-point GNN. Their method *encourages* rather than guarantees contraction via a multi-objective problem that includes the norm of the Jacobian of the update function as a quantity to be minimized. This heuristic can work in practice, but the sequence of iterates does not definitively converge, particularly if node embeddings are initialized far from the fixed point (as the norm of the Jacobian is only penalized at the fixed point). We thus do not consider this non-linear GNN to be a true fixed-point GNN. We provide more details on existing fixed-point GNN architectures in appendix A.

## 3.2 OPTIMIZATION-BASED GNNS

Optimization-based GNNs obtain node embeddings by minimizing a convex scalar-valued graph function $E_\theta : \mathcal{G} \times \mathbb{R}^{n \times k} \to \mathbb{R}$ with respect to the aggregate graph node embeddings $\boldsymbol{H}$, where $\theta$ are parameters:

$$\boldsymbol{H}^* = \arg\min_{\boldsymbol{H}} E_\theta(\mathcal{G}, \boldsymbol{H}). \tag{4}$$

Assuming $E_\theta$ is separable per node, $E_\theta$ can be expressed as:

$$E_\theta(\mathcal{G}, \boldsymbol{H}) = \sum_{i=1}^n e_\theta \left(\boldsymbol{h}_i, \boldsymbol{x}_i, \{\boldsymbol{h}_j, \boldsymbol{x}_j, \boldsymbol{e}_{ij} \mid j \in \mathcal{N}(i)\}\right) = \sum_{i=1}^n e_\theta^i. \tag{5}$$

Crucially, due to the dependence of each $e_\theta^i$ on only local information, gradient-based minimization of $E_\theta$ can be expressed per node via message passing:

$$\boldsymbol{h}_i(t+1) = \boldsymbol{h}_i(t) - \alpha \sum_{j \in \mathcal{N}(i) \cup \{i\}} \boldsymbol{g}_{ji}(t) \tag{6}$$

$$\boldsymbol{g}_{ji}(t) := \nabla_{\boldsymbol{h}_i} e_\theta^j(\boldsymbol{h}_j(t), \{\boldsymbol{h}_{j'}(t) | j' \in \mathcal{N}(j)\}), \tag{7}$$

where $\alpha \in \mathbb{R}_{>0}$, and node and edge features are omitted for clarity. We assume that at time $t$, node $i$ obtains the values $\boldsymbol{g}_{ji}$ and $\boldsymbol{h}_j$ (needed to compute $\boldsymbol{g}_{ii}$) from neighbors $j \in \mathcal{N}(i)$. The number of iterations is dictated by the convergence of the embeddings to a fixed point.

Existing optimization-based GNNs (Yang et al., 2021; Zhu et al., 2021; Ma et al., 2021) use an objective $E_\theta$ where node embeddings arise as:

$$\boldsymbol{H}^* = \arg\min_{\boldsymbol{H}} \gamma ||\boldsymbol{H} - g_\theta(\boldsymbol{X})||_F^2 + \beta \mathrm{tr}(\boldsymbol{H}^T \boldsymbol{L} \boldsymbol{H}), \tag{8}$$

where $g_\theta : \mathbb{R}^p \to \mathbb{R}^k$ and is applied independently to each node feature, $\boldsymbol{L}$ is a function of $\boldsymbol{A}$ and can be viewed as a generalized incidence matrix (assumed to be symmetric and positive semi-definite), $\gamma$ and $\beta$ are constants, and $\mathrm{tr}()$ is the trace. The first term drives node embeddings to approximate some function of the node features $\boldsymbol{X}$, and the second term is a regularizer that rewards smoothness of neighboring node embeddings. The overall objective $E_\theta$ is separable per node. We refer to optimization-based GNNs which use this form of objective as GSDGNNs, since minimizing the objective can be interpreted as performing graph signal denoising (GSD).

Previous work has shown that embeddings obtained by the fixed-point message passing scheme defined by APPNP (Gasteiger et al., 2019) and EIGNN (Liu et al., 2021) correspond to minimization of this form of objective (Ma et al., 2021; Yang et al., 2021). Correspondence between fixed-point GNNs and optimization-based GNNs does not always exist; while node embeddings in optimization-based GNNs can always be expressed as satisfying a fixed-point equation (i.e., the gradient of the convex graph function is equal to zero at the solution), not all fixed-point equations correspond with convex optimization problems. For this reason, we distinguish between these two types of implicit GNNs.

## 4 ASYNCHRONOUS GNN INFERENCE

In this section, we discuss GNN inference under the asynchronous execution model presented in Section 2.1. In the general asynchronous update in Equation (2) it is assumed that node $i$ has access to all of the values $\boldsymbol{h}_1, ... \boldsymbol{h}_n$ required for performing its update. In general these values could be obtained, for example, by providing nodes access to a shared memory structure which contains these values. However, in this work, we are interested the setting where neither centralized memory nor a centralized controller are used. We assume that individual nodes store their own values $\boldsymbol{h}_i$ (and $\boldsymbol{x}_j$, $\boldsymbol{e}_{ij}$), and if these values are needed by other nodes to perform their update, they must obtain them by communicating with neighbors. Our aim in this section is to demonstrate that node updates can be performed *only* using information that can be obtained through local communication.

### 4.1 EXPLICITLY-DEFINED GNN AND FIXED-POINT GNN INFERENCE UNDER PARTIAL ASYNCHRONISM

Without loss of generality, assume that the embedding dimension $k$ is fixed for all layers of parameterized node update functions. We do not write $f_\theta$ indexed by layer, but this is straightforwardly generalized to the case of layer-specific parameters and functions described in Section 2.2. As a slight abuse of notation, let $f_\theta^i$ denote $f_\theta$ applied to node $i$'s neighborhood. Without loss of generality, assume the embedding update function $f_\theta$ is continuously differentiable, so that the following restriction can be stated:

$$j \notin \mathcal{N}(i) \implies \frac{\partial f_\theta^i}{\partial \boldsymbol{h}_j}(\boldsymbol{z}) = 0 \quad \forall \boldsymbol{z} \in \mathbb{R}^k. \tag{9}$$

With this restriction, the general node update from Equation (2) can be adapted to describe partially asynchronous message passing, in which node updates are performed using only information from a node's local neighborhood. In particular, for node $i$ and for $t \geq 0$ and $t \in T^i$, the update equation is:

$$\boldsymbol{h}_i(t+1) = f_\theta(\boldsymbol{h}_i(t), \{\boldsymbol{h}_j(\tau_j^i(t)) \mid j \in \mathcal{N}(i)\}) \tag{10}$$

where we omit node and edge features for clarity. Note the crucial difference introduced by asynchrony: the neighbor data $\boldsymbol{h}_j(\tau_j^i(t))$ may correspond to the incorrect layer in the network. For explicitly-defined GNNs such as GCN or GAT, the number of iterations $|T_i|$ executed by each node is fixed and equal to the number of layers $L$ in the GNN. For fixed point GNNs, the number of iterations is not pre-specified.

Since explicitly-defined GNNs implement a specific feed-forward neural network architecture, inference using Equation (10) corrupts the computation performed by the network. We illustrate this in Figure 1, where asynchrony results in a (different) computation graph with some connections removed, and new connections that are not present in the original synchronous computation graph. This means that there are no convergence guarantees under partial asynchrony, and in general the final node embeddings may vary significantly with respect to the particular node update sequence.

In contrast, fixed-point GNNs in which $F_\theta$ is contractive with respect to embeddings $\boldsymbol{h}$ are provably robust to partially asynchronous inference. In particular, they satisfy the assumptions of the following proposition (Bertsekas, 1983).

**Proposition 4.1.** *If $F_\theta : \mathcal{G} \times \mathbb{R}^{nk} \to \mathbb{R}^{nk}$ is contractive with respect to node embeddings $\boldsymbol{h}$, then under the bounded staleness conditions in Theorem 2.1, the fixed-point iteration of Equation 10 converges.*

## 4.2 Optimization-based GNN Inference under Partial Asynchrony

In order to examine inference of optimization-based GNNs under partial asynchrony, we assume gradient-based optimization is used in computing node embeddings. Recall that optimization-based GNNs are performing a minimization of a separable objective $E_\theta$, as defined in Equation (5). We thus state the following restriction, analogous to the restriction in Equation (9):

$$j \notin \mathcal{N}(i) \implies \frac{\partial e_\theta^i}{\partial \boldsymbol{h}_j}(\boldsymbol{z}) = 0 \text{ for all } \boldsymbol{z} \in \mathbb{R}^k. \tag{11}$$

We could then naively adapt the general node update equations from Equation (1) to describe partially asynchronous, gradient-based minimization of $E_\theta$ as follows:

$$\boldsymbol{h}_i(t+1) = \boldsymbol{h}_i(t) - \alpha \sum_{j \in \mathcal{N}(i) \cup \{i\}} \boldsymbol{g}_{ji}(\tau_j^i(t)) \tag{12}$$

$$\boldsymbol{g}_{ji}(\tau_j^i(t)) := \nabla_{\boldsymbol{h}_i} e_\theta^j(\boldsymbol{h}_j(\tau_j^i(t)), \{\boldsymbol{h}_{j'}(\tau_{j'}^i(t)) | j' \in \mathcal{N}(j)\}), \tag{13}$$

where $\alpha \in \mathbb{R}_{>0}$ is the step size, and node and edge features are omitted for clarity. This formulation would allow us to directly cite a proof of convergence from Bertsekas and Tsitsiklis (1989), as we did in Section 4.1 for partially asynchronous fixed-point GNN inference.

However, recall that for a node to perform an update using only local communication, we previously assumed that $\boldsymbol{g}_{ji}$ was obtained by node $i$ from its neighbor $j$. With the formulation in Equations 12 and 13, node $j$ cannot provide node $i$ with $\boldsymbol{g}_{ji}$ since the value depends on node $i$'s view of the embeddings (and features) of the neighbors of node $j$, rather than deferring to node $j$'s view of its neighbors. That is, node $i$ needs access to information about its 2-hop neighbors in addition to its 1-hop neighbors. Since we assume communication is only possible with 1-hop neighbors, 2-hop neighbor information would need to be forwarded by direct neighbors of node $i$. This inflates the cost of communication, requiring a number of bits per transmission which is proportional to $|\mathcal{N}(j)|$. Furthermore, as the number of neighbors that are shared between node $i$ and node $j$ increases, the contents of these messages become increasingly redundant.

Instead, we preserve fully local communication and fixed-size transmissions where neighbors $j$ send fixed-size packets containing only $(\boldsymbol{h}_j, \boldsymbol{g}_{ji})$ to node $i$ by defining $\boldsymbol{g}_{ji}(\tau_j^i(t))$ as follows:

$$\boldsymbol{g}_{ji}(\tau_j^i(t)) := \nabla_{\boldsymbol{h}_i} e_\theta^j(\boldsymbol{h}_j(\tau_j^i(t)), \{\boldsymbol{h}_{j'}(\tau_{j'}^j(\tau_j^i(t))) : j' \in \mathcal{N}(j)\}). \tag{14}$$

The crucial difference between equations Equation (13) and Equation (14) is that instead of $\boldsymbol{g}_{ji}(\tau_j^i(t))$ depending on node $i$'s view of 2-hop neighbors $j'$ at time $t$, it now depends on neighbor $j$'s view of its neighbors at time $\tau_j^i(t)$, the time corresponding to node $i$'s view of $j$ at time $t$.

**Proposition 4.2.** *If $E_\theta$ is strongly convex and separable per node, and is twice differentiable (w.r.t. $\boldsymbol{H}$) with a Hessian of bounded norm, then for a sufficiently small step size and the bounded staleness conditions in Theorem 2.1, the optimization procedure of Equations 12 and 14 will converge when executed under partial asynchrony.*

See Appendix D for a proof adapting results from Bertsekas and Tsitsiklis (1989).

## 5 ENERGY GNNS

Under the assumptions of Section 2.1, implicitly-defined GNNs in which node embeddings are updated iteratively using local information are well suited for partially asynchronous, decentralized, and distributed inference. However, the diversity of existing implicit GNN architectures is limited (see Section 3).

We propose a novel implicitly-defined, optimization-based GNN architecture which we call the *energy GNN*. Energy GNNs compute node embeddings that minimize a parameterized, convex graph function $E_\theta$, which we refer to as the 'energy' function. In contrast to previous work on optimization-based GNNs, our energy function makes use of partially input-convex neural networks (PICNNs, Amos et al. (2017)). PICNNs are scalar-valued neural networks that constrain the parameters in such a way that the network is convex with respect to a specified subset of the inputs. This exposes a rich and flexible class of convex energy functions $E_\theta$ of the form:

$$E_\theta(\mathcal{G}, \boldsymbol{H}) = \sum_{i=1}^{n} e_\theta^i \qquad \text{where} \tag{15}$$

$$e_\theta^i = u\left(\boldsymbol{m}_i, \boldsymbol{h}_i, \boldsymbol{x}_i; \theta_u\right) + (\beta/2)||\boldsymbol{h}_i||_2^2 \tag{16}$$

$$\boldsymbol{m}_i = \sum_{j \in \mathcal{N}(i)} m(\boldsymbol{h}_i, \boldsymbol{h}_j, \boldsymbol{x}_i, \boldsymbol{x}_j, \boldsymbol{e}_{ij}; \theta_m) \tag{17}$$

where $m$ is a function that is both convex and nondecreasing (in each dimension) with respect to $\boldsymbol{h}_j$ and $\boldsymbol{h}_i$, and the function $u$ is convex with respect to $\boldsymbol{m}_i$ and $\boldsymbol{h}_i$. These functions are both implemented as PICNNs and their composition is convex. Summing these functions along with the squared norm penalty results in $E_\theta$ being strongly convex with respect to the node embeddings $\boldsymbol{H}$. This architecture for the energy can be described as a (single layer) partially input-convex GNN; we provide more details in Appendix B.

This formulation for $E_\theta$ offers significantly more flexibility than the architectures of other implicitly-defined GNNs. The functions $m$ and $u$ are parameterized by multi-layer PICNNs, and any combination of inputs to $m$ and $u$ are valid provided that $E_\theta$ remains a convex function of $\boldsymbol{H}$. This means that edge features are easily incorporated into the model, and neighbor-specific or neighbor agnostic messages can be used (e.g., $m$ can take in information from just a node's neighbor, or information pertaining to both a node and its neighbor); in our experiments in Section 6.3 we show that this translates empirically to improved performance on various tasks. Since the aggregation in Equation (17) is only constrained to be a non-negative sum over neighbors, it can be replaced with, for example, a mean or a sum weighted by the entries of the symmetric renormalized adjacency matrix (as in GCNs, see Appendix A). Alternatively, Equation (17) can easily incorporate a neighbor attention mechanism (as in GATs), where neighbor contributions to the sum are scaled by neighbor-specific attention weights (see Appendix B). The attention weights can depend on any of the non-convex inputs (i.e., the features).

## 6 EXPERIMENTS

### 6.1 SYNTHETIC MULTI-AGENT TASKS

We perform experiments on several synthetic datasets, motivated by prediction tasks which are of interest for multi-agent systems where distributed, asynchronous inference is desirable. We describe each task and associated dataset below.

**Chains** The ability to communicate information across long distances in a group of agents is important when agent predictions depend on global information. This communication is made more difficult in the absence of a central controller (as is the case for distributed, asynchronous inference). The chains dataset, used in Gu et al. (2020); Liu et al. (2021), is meant to evaluate the ability to capture long-range dependencies between nodes. The dataset consists of $p$ undirected linear graphs with $l$ nodes, with each graph having a label $k \in \{1, ..., p\}$. The task is node classification of the graph label, where class information is contained *only* in the feature of the first node in the chain; the node feature matrix $\boldsymbol{X} \in \mathbb{R}^{n \times p}$ for a graph with class $k$ has $\boldsymbol{X}_{1,k} = 1$ and zeros at all other indices. Perfect classification accuracy indicates that information is successfully propagated from the first node to the final node in the chain. For our dataset, we use chains with length $l = 100$.

**Counting** Counting the number of agents in a group may be important in various multi-agent tasks. This value can be used, for example, to calculate means, or in agent decisions which rely on group size. We construct a dataset meant to evaluate the ability of GNNs to count in undirected chain graphs. Our dataset consists of 50 graphs with 1-50 nodes. Since no informative node features are present for this task, we set

node features as one-hot embeddings of node degrees. The prediction target for each node in a given graph is the total number of nodes in that graph.

**Sums**  For this task, we consider summation, a basic functional building block relevant for many multi-agent tasks. For instance, in reinforcement learning tasks, agents might aim to perform actions that optimize their collective rather than individual rewards, requiring each agent to sum the rewards associated with all other agents. Many distributed and asynchronous algorithms exist for summation (Kempe et al., 2003). We construct a dataset to evaluate the ability of GNNs to perform binary sums in undirected chain graphs. Our data are 2000 graphs with 50 nodes each, with different instantiations of binary node features $x_i \in \{0, 1\}$. The prediction target for each node in a given graph is $\hat{y}_i := \sum_i x_i$.

**Coordinates**  A common task for multi-agent collectives such as robot swarms is localization. This problem has previously been tackled in various ways that all employ a bespoke algorithm tailored for the task (Todescato et al., 2016; Huang and Tian, 2017; 2018). We test the ability of GNNs to solve this problem on static graphs. We construct a dataset where each node has a position in $\mathbb{R}^2$ and neighbors within some radius are connected by an edge. We do not assume a global coordinate system; instead, we focus on relative localization, where pairwise distances between nodes are maintained. Each node predicts a position in $\mathbb{R}^2$, and the objective is the mean squared error between true pairwise node distances, and distances between their predicted positions. In order to break symmetries, each node has a unique ID which is one-hot encoded and used as the node feature. Distances to connected neighbors are provided as edge features. We generate 1500 random graphs where all graphs consist of 20 nodes. We sample uniformly in the unit square to get node positions and connect nodes by an edge if they are within a distance of 0.5.

**MNIST "Terrain"**  The final synthetic task we consider is "terrain" classification. Suppose a number of agents are placed in an environment where each agent performs some local measurement, and the agents must collectively make predictions about some global state of the environment using only local communication. For this experiment, we use MNIST images (those with 0/1 labels only) to represent the environment (LeCun et al., 2010). Agents are placed at random locations in the image, and use the coordinates and pixel value at their location as their node features. The prediction target for each agent is the image label. We resize the images to $10 \times 10$ pixels, and sample 10 random pixels for agent locations. Nodes share an edge if they are within 5 pixels of each other. Unlike the previous synthetic tasks, in which existing distributed algorithms can be applied, no bespoke algorithm exists for the MNIST terrain task. This is precisely the type of problem which motivates the development of GNNs which are robust to asynchronous and distributed inference.

## 6.2 EXPERIMENTAL SETUP

For each synthetic task, we compare performance of energy GNNs to other implicitly-defined GNNs we identified in Section 3. We employ three energy GNN architecture variants; node-wise, where messages are constructed using information from individual nodes, edge-wise, where messages are constructed using information pertaining to both nodes on an edge, including edge features, and edge-wise energy GNN with neighborhood attention. In terms of other implicitly-defined GNNs, we focus on the fixed-point GNN architecture IGNN defined by Gu et al. (2020) which is described in Appendix A, and against GSDGNN, an optimization-based GNN which uses the objective from Equation (8). Two fixed point GNN architectures identified in Section 3 are excluded: EIGNN (Liu et al., 2021) and the GNN proposed by Scarselli et al. (2009). The former is excluded because the fixed point is solved for directly in the forward pass using global information rather than iteratively using local information; the latter is excluded because fixed point convergence may not be achieved. In addition to implicitly-defined GNNs, we also compare against two common explicitly-defined GNN architectures; GCN (Kipf and Welling, 2017), and GAT (Veličković et al., 2018).

The cost of the forward and backward pass (in terms of computation and/or memory) for implicitly-defined GNNs is variable, depending on the number of iterations required for convergence. We mitigate this cost during training in two ways. Since convergence for both fixed-point GNNs and optimization-based GNNs is guaranteed implicit differentiation can be used to obtain gradients of the task-specific loss function $\mathcal{L}$ with respect to parameters $\theta$. This avoids unrolling the fixed-point iterations in the backward pass, and requires a fixed amount of computation and memory. We derive the gradient in Appendix C. Furthermore, since the solution of the forward pass is unique (i.e. not dependent on the initialization of $\boldsymbol{H}$), the number of iterations in the forward pass can be reduced by initializing $\boldsymbol{H}$ to be the solution from the previous epoch of training.

In our work we employ both of these strategies during training. Additional training details for the synthetic experiments are in Appendix F.

We additionally perform experiments on benchmark datasets MUTAG (Srinivasan et al., 1996), PROTEINS (Borgwardt et al., 2005), and PPI (Hamilton et al., 2017) for node and graph classification to evaluate energy GNNs as a synchronous GNN architecture. Although our objective is not performance in the synchronous setting, we show that they are nevertheless competitive on each dataset. Details are provided in Appendix G.

## 6.3 RESULTS

In Table 1, we report synchronous performance of each GNN architecture on the synthetic tasks. For regression tasks (counting, sums, coordinates) task performance is calculated as the root mean squared error over the test dataset normalized by the root mean value of the test dataset prediction targets. For classification tasks (chains, MNIST) task performance is calculated as the mean test dataset classification error. Table 1 reports performance for each task, with mean and standard deviation taken across 10 dataset folds and 5 random parameter seeds.

The results of the synthetic experiments empirically demonstrate the superiority of our energy GNN architecture compared to other implicitly-defined GNNs. The node-wise energy GNN architecture improves performance over IGNN and GSDGNN, which we attribute to the use of PICNNs. When edge-wise rather than node-wise information is used in constructing messages to neighbors, further improvements in performance are observed. The strong performance on tasks requiring long-distance communication between nodes for correct predictions (chains, counting, and sums) shows that our architecture is capable of capturing long-range dependencies between node predictions.

Table 1: Task performance on test data, reported as percentage error (relative root mean squared error for COUNT, SUM, COORDINATES). Mean and standard deviation are across 10 random seeds and 5 train/test splits. Although inference on these experiments is done synchronously, the poor performance of the explicitly-defined GCN and GAT can be attributed to their depth-limited ability to propagate information.

| MODEL | CHAINS | COUNT | SUM | COORDINATES | MNIST |
|---|---|---|---|---|---|
| IGNN | $26.9 \pm 13.3$ | $40.2 \pm 5.5$ | $12.8 \pm 0.8$ | $52.0 \pm 5.5$ | $30.4 \pm 0.7$ |
| GSDGNN | $35.9 \pm 6.3$ | $40.3 \pm 0.6$ | $13.3 \pm 0.2$ | $44.0 \pm 1.0$ | $29.3 \pm 0.6$ |
| ENERGY GNN NODE-WISE | $15.8 \pm 17.9$ | $19.5 \pm 1.7$ | $12.2 \pm 0.7$ | $41.3 \pm 2.4$ | $13.8 \pm 0.8$ |
| ENERGY GNN EDGE-WISE | $1.2 \pm 2.2$ | $4.0 \pm 3.6$ | $\mathbf{4.9 \pm 3.3}$ | $33.5 \pm 3.2$ | $\mathbf{13.0 \pm 0.6}$ |
| ENERGY GNN + ATTENTION | $\mathbf{0.25 \pm 0.5}$ | $\mathbf{3.6 \pm 3.8}$ | $6.0 \pm 4.0$ | $\mathbf{30.9 \pm 1.8}$ | $13.8 \pm 0.8$ |
| GCN | $47.0 \pm 0.0$ | $40.7 \pm 1.0$ | $13.1 \pm 0.3$ | $53.2 \pm 0.9$ | $29.7 \pm 0.5$ |
| GAT | $47.0 \pm 0.0$ | $41.5 \pm 0.8$ | $12.9 \pm 0.5$ | $39.3 \pm 1.0$ | $15.3 \pm 3.2$ |

Table 2: Decrease in task performance (decrease in accuracy for CHAINS, MNIST, and increase in relative RMSE for COUNT, SUM, COORDINATES) observed from switching from synchronous to asynchronous inference on sub-sample of test data (10 samples) using one trained model instance. Mean and standard deviation are across 5 asynchronous runs. The poor performance of GCN and GAT are consistent with the expected unreliability of explicitly-defined GNNs with asynchronous inference. Decreases in task performance for all implicitly-defined GNNs (IGNN, GSDGNN, and energy GNN variants) is less than $0.1\%$ (i.e. result from numerical error); these values are omitted from the table for concision.

| MODEL | CHAINS | COUNT | SUM | COORDINATES | MNIST |
|---|---|---|---|---|---|
| GCN | $38.8 \pm 3.3$ | $584.6 \pm 42.4$ | $2.6 \pm 0.2$ | $63.4 \pm 0.0$ | $37.4 \pm 10.3$ |
| GAT | $6.6 \pm 1.8$ | $250.3 \pm 58.6$ | $45.1 \pm 2.1$ | $97.0 \pm 34.4$ | $50.4 \pm 3.5$ |

In Table 2, we demonstrate empirically that explicitly-defined architectures such as GCN and GAT perform poorly and unreliably under asynchrony, with task performance decreasing for all experiments. In our experiments, we simulate asynchronous inference; our algorithm is in Appendix E. In most cases the variance of the performance decrease is large as a result of inconsistent predictions under different random node update schedules and communication delays. In cases where the variance is low, we observe that predictions

for different random schedules collapse to the same/similar values due to non-linearities in the architecture. For implicitly-defined GNNs, the decrease in performance under asynchronous inference is less than 0.1%, empirically confirming the convergence guarantees given by Proposition 4.1 and Proposition 4.2.

## 7 RELATED WORK

Dudzik et al. (2023) axiomatically derive GNN layers which are invariant to asynchrony. For instance, using max-aggregation for messages is by construction invariant to asynchrony; messages from neighbors can be processed as they arrive rather than waiting for all inputs to be available, and the final aggregated message will still be correct. Later work in this thread aims at enforcing asyncrony invariance through a self-supervised loss function rather than using layers which are asynchrony invariant by construction (Monteagudo-Lago et al., 2024). The motivation for this line of work is to apply GNNs for neural algorithmic reasoning, where the GNN is trained to emulate classical algorithms such as the Bellman-Ford algorithm. Enforcing asyncrony invariance in the GNN serves to improve alignment between the algorithm to be learned and the GNN computation. The asynchronous model of computation used in these works differs from ours in that nodes must maintain information from all layers in the GNN, rather than executing layer-wise.

Faber and Wattenhofer (2024) also explore asynchronous communication between nodes during GNN inference. Their approach consists of processing node updates in a queue, rather than applying all node updates in a layer in parallel. The goal of their work is to leverage sequential node updates to improve under-reaching and over-squashing in learned node embeddings. Asynchronous execution in their model is deterministic in that there are no random delays in communication between nodes; nodes receive messages and perform updates in the same order across different runs. They show that random delays and node update orderings leads to unstable training and results in decreased task performance. This makes their approach unsuitable for the settings we consider, in which asynchrony is not controlled.

## 8 CONCLUSION

GNNs have the potential to provide learning frameworks for decentralized multi-agent systems, with applications to robotics, remote sensing, and other domains. However, asynchronous execution and unreliable communication between agents are common features in real-world deployment, and conventional GNN architectures do not generate reliable predictions in this setting. In this work, we characterize the class of implicitly-defined GNNs as being provably robust to partially asynchronous inference. Motivated by lack of diversity in this class of architectures, we contribute a novel addition in the form of energy GNNs, which achieve better performance than other implicitly-defined GNNs on a number of synthetic multi-agent tasks.

The positive results of our synthetic experiments motivates additional work in applying GNN architectures to multi-agent systems, particularly making use of implicitly-defined GNNs to generate state representations for control tasks, rather than for prediction tasks. A specific line of work which we expect to be interesting is real-time inference on *dynamic* graphs because of the relevance to problems in, e.g., robotics. Distributed and asynchronous inference for datasets consisting of large graphs, where inference must be distributed among multiple processors due to scale, is another application for implicitly-defined GNNs which should be explored.

There are several limitations of this work, particularly in training implicitly-defined GNNs. The forward pass requires convergence of node embeddings via a fixed point iteration, and requires an amount of time/computation that cannot be known in advance. During training, we decrease the number of iterations required for convergence by initializing node embeddings to those from the previous epoch of training. Real-time inference in multi-agent systems is likely to also enjoy the benefits of a warm-start initialization. This is because the graph features and structure are often expected to change slowly over time, meaning that after an initial "boot-up" from random initialization, a continuously operating implicitly-defined GNN is likely to have embeddings initialized close to the solution from one time step to the next (assuming the node embeddings $\boldsymbol{H}*$ are not extremely sensitive to small changes in the graph). For optimization-based GNNs using gradient-based optimization in the forward pass, the condition number of the Hessian of $E_\theta$ affects the convergence rate, but is difficult to control. Similarly, if $F_\theta$ is poorly conditioned in fixed-point GNNs, convergence can be slow and is susceptible to numerical instability. During training, we use implicit differentiation to obtain gradients w.r.t. parameters rather than simple backpropagation; this avoids unrolling through the iterative process in the forward pass, but requires solving a linear system using the Hessian of $E_\theta$ (for optimization-based GNNs) or the Jacobian of $F_\theta$ (for fixed-point GNNs), either of which may be poorly conditioned. Finally, operationalizing the theoretical results may require constants that are not readily available.

ACKNOWLEDGMENTS

This work was partially supported by NSF grants IIS-2007278 and OAC-2118201.

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

## A  GNN ARCHITECTURES

**Graph Convolutional Networks**  GCNs (Kipf and Welling, 2017) replace the adjacency matrix $\boldsymbol{A}$ with the symmetric normalized adjacency matrix with added self-loops, $\tilde{\boldsymbol{A}} = (\boldsymbol{D} + \boldsymbol{I})^{-\frac{1}{2}}(\boldsymbol{A} + \boldsymbol{I})(\boldsymbol{D} + \boldsymbol{I})^{-\frac{1}{2}}$. With node embeddings initialized to be equal to the node features, $f_\theta^\ell$ is defined as:

$$\boldsymbol{m}_i^\ell := \sum_{j \in \mathcal{N}(i)} \tilde{\boldsymbol{A}}_{i,j} \theta_m^\ell \boldsymbol{h}_j^\ell \qquad\qquad \boldsymbol{h}_i^{\ell+1} := \mathrm{ReLU}(\boldsymbol{m}_i^\ell), \qquad (18)$$

where $\theta_m^\ell \in \mathbb{R}^{k^{(\ell)} \times k^{(\ell)}}$.  This update can be succinctly described at the graph level as $\boldsymbol{H}^{\ell+1} = \mathrm{ReLU}(\tilde{\boldsymbol{A}} \boldsymbol{H}^\ell \boldsymbol{W}^\ell)$.  Note that for explicitly-defined message passing GNNs which have $L$ layers, such as GCN, it is impossible to propagate information farther than $L$ hops.

**Graph Attention Networks**  GATs (Veličković et al., 2018) apply an attention mechanism to determine the weighting of information from different neighbors. With node embeddings initialized to be equal to the node features, $f_\theta^\ell$ is defined as:

$$\alpha_{i,j} = \frac{\exp\left(\mathrm{LeakyReLU}((\theta_a^\ell)^T [\theta_m^\ell \boldsymbol{h}_i || \theta_m^\ell \boldsymbol{h}_j])\right)}{\sum_{k \in \mathcal{N}(i)} \exp\left(\mathrm{LeakyReLU}((\theta_a^\ell)^T [\theta_m^\ell \boldsymbol{h}_i || \theta_m^\ell \boldsymbol{h}_j])\right)} \qquad (19)$$

$$\boldsymbol{m}_i^\ell := \sum_{j \in \mathcal{N}(i)} \alpha_{i,j} \theta_m^\ell \boldsymbol{h}_j^\ell \qquad (20)$$

$$\boldsymbol{h}_i^{\ell+1} := \mathrm{ReLU}(\boldsymbol{m}_i^\ell), \qquad (21)$$

where $\theta_m^\ell \in \mathbb{R}^{q \times k^{(\ell)}}$, and $\theta_a^\ell \in \mathbb{R}^{2q}$ are additional parameters used in aggregation. Multiple attention heads can be used, in which there are $k$ aggregation functions with their own parameters. The messages generated by each of the attention heads are either concatenated or averaged to generate a single message $\boldsymbol{m}_i^\ell$.

**Implicit Graph Neural Networks (IGNNs)**  IGNNs Gu et al. (2020) are a fixed point GNN architecture, in which each parameterized node update layer is contractive with respect to the node embeddings.  For simplicity, we assume a single parameterized layer $f_\theta$ and exclude the layer superscript from our notation. The parameterized node embedding update is repeated for steps $t = 0, ..., T - 1$, where the stopping point $T$ is determined by when node embeddings converge (within some numerical tolerance). IGNNs use a similar embedding update function as GCN, but add node features as an additional input to the update function. A layer $f_\theta$ is defined as:

$$\boldsymbol{m}_i(t) := \sum_{j \in \mathcal{N}(i)} \tilde{\boldsymbol{A}}_{i,j} \theta_m \boldsymbol{h}_j(t) \qquad\qquad \boldsymbol{h}_i(t+1) := u(\boldsymbol{m}_i(t) + g(\boldsymbol{x}_i; \theta_u)), \qquad (22)$$

where $\theta_m \in \mathbb{R}^{k \times k}$, $g_\theta : \mathbb{R}^{n \times p} \to \mathbb{R}^{n \times k}$ and $\phi$ is a component-wise non-expansive function such as ReLU. Convergence is guaranteed by constraining $||\theta_m||_\infty < \lambda_{pf}(\tilde{\boldsymbol{A}})^{-1}$, where $\lambda_{pf}(\tilde{A})$ is the maximum eigenvalue of $|\tilde{\boldsymbol{A}}|$. This ensures that the update is contractive, a sufficient condition for convergence. Since the fixed point is unique, $\boldsymbol{h}_i(0)$ can be initialized arbitrarily (although convergence time will vary).

**Efficient Implicit Graph Neural Networks (EIGNNs)**  EIGNNs Liu et al. (2021) are another fixed point GNN architecture which are very similar to IGNNs. The message passing iteration is constructed such that a closed-form solution can be obtained for the fixed point of a layer, which is more efficient than iterating message passing to convergence. A layer $f_\theta$ is defined as:

$$\boldsymbol{m}_i(t) := \sum_{j \in \mathcal{N}(i)} \gamma \alpha \tilde{\boldsymbol{A}}_{i,j} (\theta_m)^T (\theta_m) \boldsymbol{h}_j(t) \qquad\qquad \boldsymbol{h}_i(t+1) := \boldsymbol{m}_i(t) + \boldsymbol{x}_i, \qquad (23)$$

where $\theta_m \in \mathbb{R}^{k \times k}$, $\alpha > 0$ is a scaling factor equal to $\frac{1}{||(\theta_m)^T (\theta_m)||_F + \epsilon}$ with arbitrarily small $\epsilon$, and $\gamma \in (0, 1]$ is an additional scaling factor. The overall scaling factor $\gamma\alpha$ is chosen to ensure that the update is contractive, from which it follows that the sequence of iterates converges.

**Nonlinear Fixed Point GNN**  Scarselli et al. (2009) introduce a general nonlinear fixed point GNN whose update can be written as

$$\boldsymbol{m}_i(t) = \sum_{j \in \mathcal{N}(i)} m(\boldsymbol{h}_i(t), \boldsymbol{h}_j(t), \boldsymbol{x}_i, \boldsymbol{x}_j, \boldsymbol{e}_{ij}; \theta_m) \qquad (24)$$

$$\boldsymbol{h}_i(t+1) = u(\boldsymbol{m}_i(t), \boldsymbol{h}_i(t), \boldsymbol{x}_i; \theta_u). \qquad (25)$$

where $m$ and $u$ are multi-layer neural networks. As we discuss in Section 3, this flexible parameterization of message passing comes at a cost: it is difficult to enforce that the overall update is definitely contractive.

# B  INPUT-CONVEX GNN ARCHITECTURE DETAILS

As in Amos et al. (2017), we construct a parametric family of neural networks $f_\theta(\boldsymbol{x}, \boldsymbol{y})$ with inputs $\boldsymbol{x} \in \mathbb{R}^n, \boldsymbol{y} \in \mathbb{R}^m$ which are convex with respect to $\boldsymbol{y}$ (i.e. a subset of the inputs). An $L$-layer partially convex neural network is defined by the following recurrences:

$$\boldsymbol{u}_{\ell+1} = \tilde{g}_\ell(\tilde{\boldsymbol{W}}_\ell \boldsymbol{u}_\ell + \tilde{\boldsymbol{b}}_\ell)$$
$$\boldsymbol{z}_{\ell+1} = g_\ell(\\
\boldsymbol{W}_\ell^{(z)}(\boldsymbol{z}_\ell \circ [\boldsymbol{W}_\ell^{(zu)}\boldsymbol{u}_\ell + \boldsymbol{b}_\ell^{(z)}]_+)+\\
\boldsymbol{W}_\ell^{(y)}(\boldsymbol{y} \circ (\boldsymbol{W}_\ell^{(yu)}\boldsymbol{u}_\ell + \boldsymbol{b}_\ell^{(y)}))+\\
\boldsymbol{W}_\ell^{(u)}\boldsymbol{u}_\ell + \boldsymbol{b}_\ell)$$

$$f_\theta(\boldsymbol{x}, \boldsymbol{y}) = \boldsymbol{z}_L, \quad \boldsymbol{u}_0 = \boldsymbol{x}, \quad \boldsymbol{z}_0 = \boldsymbol{0}$$

Provided the $\boldsymbol{W}^{(z)}$ are elementwise nonnegative for all layers $\ell$, and the activation functions $g_\ell$ are nondecreasing in each argument, it follows that $f_\theta$ is convex in $\boldsymbol{y}$.

In the context of an energy GNN, $E_\theta$ is comprised of two PICNNs, specialized to operate on graph structures; we refer to the resulting architecture as a partially input-convex GNN (PICGNN). The message function $m$ in Equation (17) corresponds to a PICNN which has node embeddings $\boldsymbol{h}_i, i = 1, \ldots, n$ as its convex inputs (all other features are non-convex inputs). A second PICNN corresponds with the update function $u$ in Equation (16), which is convex in the node embeddings and the messages computed by the message function, and nonconvex in the node features. The graph energy $E_\theta$ can then be written as the sum of the outputs of the second PICNN.

## B.1  NEIGHBORHOOD ATTENTION

We can incorporate a neighborhood attention mechanism in the PICGNN by modifying the aggregation of messages $\boldsymbol{m}_i$ as follows:

$$\boldsymbol{m}_i = \sum_{j \in \mathcal{N}(i)} \alpha_{i,j} m(\boldsymbol{h}_i, \boldsymbol{h}_j, \boldsymbol{x}_i, \boldsymbol{x}_j, \boldsymbol{e}_{ij}; \theta_m) \qquad \text{where} \tag{26}$$

$$\alpha_{i,j} = \frac{\exp\left(\text{LeakyReLU}(\theta_a)^T[\theta_{m_x}\boldsymbol{x}_i || \theta_{m_x}\boldsymbol{x}_j || \theta_{m_e}\boldsymbol{e}_{ij}])\right)}{\sum_{k \in \mathcal{N}(i)} \exp\left(\text{LeakyReLU}((\theta_a)^T[\theta_{m_x}\boldsymbol{x}_i || \theta_{m_x}\boldsymbol{x}_j || \theta_{m_e}\boldsymbol{e}_{ij}])\right)} \tag{27}$$

where $\theta_{m_x} \in \mathbb{R}^{q \times p}$, $\theta_{m_e} \in \mathbb{R}^{s \times r}$ and $\theta_a \in \mathbb{R}^{2q+s}$ are additional parameters used in aggregation. As in GATs, multiple attention heads can be used, in which there are $k$ aggregation functions with their own parameters. The messages generated by each of the attention heads are either concatenated or averaged to generate a single message $\boldsymbol{m}_i$.

# C  IMPLICIT DIFFERENTIATION

Since we use an optimization procedure to compute the node embeddings within the forward pass in implicitly-defined GNNs, we need to obtain derivatives of the node embeddings with respect to the parameters of the model. We compute derivatives by implicitly differentiating the optimality conditions. For fixed-point GNNs, we use the fact that at the fixed point, we have parameters $\theta^* \in \mathbb{R}^p$ and node embeddings $\boldsymbol{H}^* \in \mathbb{R}^{n \times k}$ such that:

$$g(\boldsymbol{H}^*, \theta) = f_\theta(\boldsymbol{H}^*) - \boldsymbol{H}^* = \boldsymbol{0}. \tag{28}$$

For optimization-based GNNs, we use the fact that at the solution of the minimization problem, we have:

$$g(\boldsymbol{H}^*, \theta) = \frac{\partial E_{\theta^*}}{\partial \boldsymbol{H}}(\boldsymbol{H}^*) = \boldsymbol{0}. \tag{29}$$

Let $h^*(\theta^*) = \boldsymbol{H}^*$, so that we can write the optimality conditions in terms of the parameters only:

$$g(h^*(\theta^*), \theta^*) = \boldsymbol{0}. \tag{30}$$

Given an objective $\mathcal{L} : \mathbb{R}^p \mapsto \mathbb{R}$, the desired quantity is the total derivative of $\mathcal{L}$ with respect to the parameters. By the chain rule,

$$\frac{d\mathcal{L}}{d\theta} = \frac{\partial\mathcal{L}}{\partial h^*}\frac{dh^*}{d\theta} + \frac{\partial\mathcal{L}}{\partial\theta}. \tag{31}$$

We compute $\frac{\partial\mathcal{L}}{\partial h^*}$ and $\frac{\partial\mathcal{L}}{\partial\theta}$ using normal automatic differentiation, and the solution Jacobian $\frac{dh^*}{d\theta}$ using implicit differentiation. Notice that, at the fixed point (where the optimality constraint is satisfied), we have:

$$\frac{d}{d\theta}g(h^*(\theta),\theta) = \mathbf{0} \tag{32}$$

$$\frac{\partial g}{\partial h^*}\frac{dh^*}{d\theta} + \frac{\partial g}{\partial\theta} = \mathbf{0} \tag{33}$$

$$\frac{\partial g}{\partial h^*}\frac{dh^*}{d\theta} = -\frac{\partial g}{\partial\theta}. \tag{34}$$

This is the primal (or tangent) system associated with the constraint function $g$. In our setup we utilize reverse mode automatic differentiation, since $p \gg 1$ parameters are mapped to a single scalar objective. Provided $\frac{\partial g}{\partial h^*}$ is invertible, we can rewrite the solution Jacobian as:

$$\frac{dh^*}{d\theta} = -\left(\frac{\partial g}{\partial h^*}\right)^{-1}\frac{\partial g}{\partial\theta}, \tag{35}$$

and substitute this expression into Equation (31) as follows:

$$\frac{d\mathcal{L}}{d\theta} = -\frac{\partial\mathcal{L}}{\partial h^*}\left(\frac{\partial g}{\partial h^*}\right)^{-1}\frac{\partial g}{\partial\theta} + \frac{\partial\mathcal{L}}{\partial\theta}. \tag{36}$$

For reverse mode, we compute the dual (or adjoint) of this equation,

$$\frac{d\mathcal{L}}{d\theta}^T = -\frac{\partial g}{\partial\theta}^T\left(\frac{\partial g}{\partial h^*}\right)^{-T}\frac{\partial\mathcal{L}}{\partial h^*}^T + \frac{\partial\mathcal{L}}{\partial\theta}^T, \tag{37}$$

And solve the dual system:

$$\frac{\partial g}{\partial h^*}^T\lambda = -\frac{\partial\mathcal{L}}{\partial h^*}^T \tag{38}$$

for the dual variable $\lambda$.

## D   PROOF OF CONVERGENCE UNDER PARTIAL ASYNCHRONY

Our guarantee of convergence uses the classical result from Bertsekas and Tsitsiklis (1989, Chapter 7.5), which depends on several formal assumptions. We reproduce these assumptions here, in the notation used in the present paper, and address how they are satisfied in our problem setting. In the following, $s_i(t)$ is the "search direction" taken by node $i$, i.e., the vector used by node $i$ to take optimization step. Ideally, this would be the negative gradient $-\nabla_{h_i}E_\theta$, but partial asynchrony means it may be a vector constructed from stale information. Our goal is to show that convergence of the optimization is nevertheless guaranteed. We use $h \in \mathbb{R}^{nk}$ to denote the unrolled embeddings $H$.

**Assumption D.1 (Bertsekas and Tsitsiklis (1989) Assumption 5.1)**
   *(a) There holds $E_\theta(h) \geq 0$ for every $h \in \mathbb{R}^{nk}$.*

   *(b) (Lipschitz Continuity of $\nabla E_\theta$) The function $E_\theta$ is continuously differentiable and there exists a constant $K_1$ such that*

$$||\nabla E_\theta(h) - \nabla E_\theta(h')|| \leq K_1||h - h'||, \quad \forall h, h' \in \mathbb{R}^{nk}$$

For part *(a)*, since both convexity and absolute continuity are preserved under nonnegative summation, the graph energy $E$ given in Equation (5) is a smooth, strictly convex function. Without loss of generality, we can assume the node energies $e_\theta^i$ described in Equation (5) satisfy $e_\theta^i(h_i) \geq 0$ for all $h_i \in \mathbb{R}^{d_i}$. This follows because the $e_\theta^i$ are strictly convex, therefore the optimal value $p_i^* = \inf\{e_\theta^i(h_i)\}$ is achieved and thus $\tilde{e}_i = e_\theta^i + p_i^*$ is nonnegative. Thus, we have that the graph energy is the sum of nonnegative terms: $E_\theta(h_1, h_2, \ldots, h_n) \geq 0$ for all $(h_1, h_2, \ldots, h_n) \in \mathbb{R}^{nk}$.

For part *(b)*, the assumption of the $e_\theta^i$ having a bounded Hessian implies that their sum also has a bounded Hessian, which further implies Lipschitz continuity of the gradient of $E_\theta$.

**Assumption D.2 (Bertsekas and Tsitsiklis (1989) Assumption 5.5)**
*(a) (Block-Descent) There holds $s_i(t)^\top \nabla_{\boldsymbol{h}_i} E_\theta(\boldsymbol{h}(t)) \leq -||s_i(t)||^2/K_3$ for all $i$ and all $t \in T^i$.*

*(b) There holds $||s_i(t)|| \geq K_2 ||\nabla_{\boldsymbol{h}_i} E_\theta(\boldsymbol{h}(t))||$ for all $i$ and all $t \in T^i$.*

For part *(a)*, the situation is slightly more complex than the conventional optimization setup, as the gradients for the search direction $s_i(t)$ are being computed from potentially-outdated neighbor embeddings, rather than it being the gradients themselves that are outdated. Writing the negative search direction for node $i$ at time $t$ in terms of the update times $\tau(t)$, we have

$$\bar{s}_i(t) := -s_i(t) \tag{39}$$

$$= \nabla_{\boldsymbol{h}_i} e_\theta^i(\boldsymbol{h}_1(\tau_1^i(t)), \dots, \boldsymbol{h}_n(\tau_n^i(t))) + \sum_{j \in \mathcal{N}(i)} \nabla_{\boldsymbol{h}_i} e_\theta^j(\boldsymbol{h}_1(\tau_1^j(\tau_j^i(t))), \dots, \boldsymbol{h}_n(\tau_n^j(\tau_j^i(t)))), \tag{40}$$

where the the gradient communicated from node $j$ may have used stale versions of the embedding both for node $i$ itself and for other nodes connected to $j$. Contrast this with the "true" gradient computed at $i$ which would be computed from its current estimate of the complete state of the graph:

$$\nabla_{\boldsymbol{h}_i} E_\theta(\boldsymbol{h}(t)) = \nabla_{\boldsymbol{h}_i} e_\theta^i(\boldsymbol{h}_1(\tau_1^i(t)), \dots, \boldsymbol{h}_n(\tau_n^i(t))) + \sum_{j \in \mathcal{N}(i)} \nabla_{\boldsymbol{h}_i} e_\theta^j(\boldsymbol{h}_1(\tau_1^i(t)), \dots, \boldsymbol{h}_n(\tau_n^i(t))) . \tag{41}$$

We introduce the following notation to simplify the exposition:

$$g_{j/i}^{k/k'}(t) := \nabla_{\boldsymbol{h}_i} e_\theta^j(\boldsymbol{h}_1(\tau_1^k(\tau_j^{k'}(t))), \dots \boldsymbol{h}_n(\tau_n^k(\tau_j^{k'}(t)))) , \tag{42}$$

which can be read as "gradient of $e_\theta^j$ with respect to $\boldsymbol{h}_i$ from the perspective of node $k$ at the time corresponding to some node $k'$'s view of node $j$ at time $t$.". With this notation, we define

$$\bar{s}_i(t) = g_{i/i}^{i/i}(t) + \sum_{j \in \mathcal{N}(i)} g_{j/i}^{j/i}(t) \qquad\qquad \nabla_{\boldsymbol{h}_i} E_\theta(\boldsymbol{h}(t)) = g_{i/i}^{i/i}(t) + \sum_{j \in \mathcal{N}(i)} g_{j/i}^{i/j}(t) . \tag{43}$$

We wish to show that the inner product between $\bar{s}_i(t)$ and $\nabla_{\boldsymbol{h}_i} E_\theta(\boldsymbol{h}(t))$ is greater than $||s_i(t)||^2/K_3$, for some $K_3 > 0$ and all $i$ and $t$.

**Lemma D.3**
*There exists an $\alpha > 0$ such that $\bar{s}_i(t)^\top \nabla_{\boldsymbol{h}_i} E_\theta(\boldsymbol{h}(t)) \geq ||s_i(t)||^2/K_3$.*

*Proof.* Starting with the squared error in the negative search direction:

$$||\bar{s}_i(t) - \nabla_{\boldsymbol{h}_i} E_\theta(\boldsymbol{h}(t))||_2^2 = (\bar{s}_i(t) - \nabla_{\boldsymbol{h}_i} E_\theta(\boldsymbol{h}(t)))^\top (\bar{s}_i(t) - \nabla_{\boldsymbol{h}_i} E_\theta(\boldsymbol{h}(t))) \tag{44}$$

$$= ||\bar{s}_i(t)||_2^2 + ||\nabla_{\boldsymbol{h}_i} E_\theta(\boldsymbol{h}(t))||_2^2 - 2\bar{s}_i(t)^\top \nabla_{\boldsymbol{h}_i} E_\theta(\boldsymbol{h}(t)) \tag{45}$$

we find an expression for the inner product:

$$\bar{s}_i(t)^\top \nabla_{\boldsymbol{h}_i} E_\theta(\boldsymbol{h}(t)) = \frac{1}{2} \left( ||\bar{s}_i(t)||_2^2 + ||\nabla_{\boldsymbol{h}_i} E_\theta(\boldsymbol{h}(t))||_2^2 - ||\bar{s}_i(t) - \nabla_{\boldsymbol{h}_i} E_\theta(\boldsymbol{h}(t))||_2^2 \right) . \tag{46}$$

and so we require the following to be greater than or equal to zero:

$$\bar{s}_i(t)^\top \nabla_{\boldsymbol{h}_i} E_\theta(\boldsymbol{h}(t)) - ||\bar{s}_i(t)||^2/K_3$$
$$= \frac{1}{2} \left( (1 - \frac{2}{K_3}) ||\bar{s}_i(t)||_2^2 + ||\nabla_{\boldsymbol{h}_i} E_\theta(\boldsymbol{h}(t))||_2^2 - ||\bar{s}_i(t) - \nabla_{\boldsymbol{h}_i} E_\theta(\boldsymbol{h}(t))||_2^2 \right) \tag{47}$$

We can use the triangle inequality to find an upper bound on the term being subtracted:

$$||\bar{s}_i(t) - \nabla_{\boldsymbol{h}_i} E_\theta(\boldsymbol{h}(t))||_2 = \left\| \left( g_{i/i}^{i/i}(t) + \sum_{j \in \mathcal{N}(i)} g_{j/i}^{j/i}(t) \right) - \left( g_{i/i}^{i/i}(t) + \sum_{j \in \mathcal{N}(i)} g_{j/i}^{i/j}(t) \right) \right\|_2 \tag{48}$$

$$= \left\| \sum_{j \in \mathcal{N}(i)} g_{j/i}^{j/i}(t) - g_{j/i}^{i/j}(t) \right\|_2 \tag{49}$$

$$\leq \sum_{j \in \mathcal{N}(i)} ||g_{j/i}^{j/i}(t) - g_{j/i}^{i/j}(t)||_2 \tag{50}$$

$$= \sum_{j \in \mathcal{N}(i)} ||\nabla_{\boldsymbol{h}_i} e_\theta^j(\boldsymbol{h}_1(\tau_1^j(\tau_j^i(t))), \dots, \boldsymbol{h}_n(\tau_n^j(\tau_j^i(t)))) - \nabla_{\boldsymbol{h}_i} e_\theta^j(\boldsymbol{h}(\tau^i(t)))||_2 . \tag{51}$$

Any difference between the states $\boldsymbol{h}(\tau^j(\tau_j^i(t)))$ and $\boldsymbol{h}(\tau^i(t))$ would arise because node $i$ and node $j$ observe different staleness states of one or more of their shared neighbors $j'$; these different staleness states correspond to differences in the number of gradient steps taken by shared neighbors $j'$, as observed by node $i$ and $j$. Note that we can assume that node $i$ and $j$ agree on the values of neighbors $j'$ of either node $i$ or $j$ which are not shared between them. Assuming the norm of the gradient is bounded, i.e., $||\nabla_{\boldsymbol{h}_{j'}} e_\theta^k||_2 \leq K_0$, and the number of neighbors a node has is bounded, i.e., $|\mathcal{N}(j')| \leq n_{\max}$, then the staleness bound $B$ and the step size $\alpha$ imply

$$||\boldsymbol{h}(\tau^j(\tau_j^i(t))) - \boldsymbol{h}(\tau^i(t))||_2 \leq \alpha B_0 K_0 \qquad \text{where} \quad B_0 := 2n_{\max}|\mathcal{N}(i) \cap \mathcal{N}(j)|B, \tag{52}$$

where we assume $\mathcal{N}(k)$ includes node $k$ itself. The constant 2 in the inequality above arises because the staleness of the gradient received by node $i$ from node $j$ is outdated by at most $B$ time units, and node $j$'s view of its neighbor embeddings is also outdated by at most $B$ time units; this means the staleness of the embeddings in the gradient received by node $i$ is stale by at most $2B$ time units. The Lipschitz continuity condition then gives

$$||\nabla_{\boldsymbol{h}_i} e_\theta^j(\boldsymbol{h}(\tau^j(\tau_j^i(t)))) - \nabla_{\boldsymbol{h}_i} e_\theta^j(\boldsymbol{h}(\tau^i(t)))||_2 \leq \alpha B_0 K_0 K_1 \tag{53}$$

and therefore

$$\bar{s}_i(t)^\top \nabla_{\boldsymbol{h}_i} E_\theta(\boldsymbol{h}(t)) - ||\bar{s}_i(t)||^2 / K_3 \tag{54}$$

$$= \frac{1}{2}\left( (1 - \frac{2}{K_3})||\bar{s}_i(t)||_2^2 + ||\nabla_{\boldsymbol{h}_i} E_\theta(\boldsymbol{h}(t))||_2^2 - ||\bar{s}_i(t) - \nabla_{\boldsymbol{h}_i} E_\theta(\boldsymbol{h}(t))||_2^2 \right) \tag{55}$$

$$\geq \frac{1}{2}\left( (1 - \frac{2}{K_3})||\bar{s}_i(t)||_2^2 + ||\nabla_{\boldsymbol{h}_i} E_\theta(\boldsymbol{h}(t))||_2^2 - (\alpha|\mathcal{N}(i)|B_0 K_0 K_1)^2 \right). \tag{56}$$

We can therefore satisfy the assumption by choosing $\alpha > 0$ such that

$$\frac{(1 - \frac{2}{K_3})||s_i(t)||_2^2 + ||\nabla_{\boldsymbol{h}_i} E_\theta(\boldsymbol{h}(t))||_2^2}{(|\mathcal{N}(i)|B_0 K_0 K_1)^2} \geq \alpha^2. \tag{57}$$

$\square$

For part *(b)* we require there to exist a lower bound on the magnitude of $s_i(t)$ relative to the magnitude of the true gradient; this prevents the step from being too small.

**Lemma D.4**

*There exists an $\alpha > 0$ such that $\bar{s}_i(t)^\top \nabla_{\boldsymbol{h}_i} E_\theta(\boldsymbol{h}(t)) \geq K_2 ||\nabla_{\boldsymbol{h}_i} E_\theta(\boldsymbol{h}(t))||^2$.*

*Proof.* We can use a nearly identical argument to that done above in part *(a)*, but instead of Equation (54) we write

$$\bar{s}_i(t)^\top \nabla_{\boldsymbol{h}_i} E_\theta(\boldsymbol{h}(t)) - K_2 ||\nabla_{\boldsymbol{h}_i} E_\theta(\boldsymbol{h}(t))||^2 \tag{58}$$

$$= \frac{1}{2}\left( ||\bar{s}_i(t)||_2^2 + (1 - 2K_2)||\nabla_{\boldsymbol{h}_i} E_\theta(\boldsymbol{h}(t))||_2^2 - ||\bar{s}_i(t) - \nabla_{\boldsymbol{h}_i} E_\theta(\boldsymbol{h}(t))||_2^2 \right) \tag{59}$$

$$\geq \frac{1}{2}\left( ||\bar{s}_i(t)||_2^2 + (1 - 2K_2)||\nabla_{\boldsymbol{h}_i} E_\theta(\boldsymbol{h}(t))||_2^2 - (\alpha|\mathcal{N}(i)|B_0 K_0 K_1)^2 \right). \tag{60}$$

We can then choose $\alpha$ to be

$$\frac{||s_i(t)||_2^2 + (1 - 2K_2)||\nabla_{\boldsymbol{h}_i} E_\theta(\boldsymbol{h}(t))||_2^2}{(|\mathcal{N}(i)|B_0 K_0 K_1)^2} \geq \alpha^2. \tag{61}$$

$\square$

**Lemma D.5**

*If $\bar{s}_i(t)^\top \nabla_{\boldsymbol{h}_i} E_\theta(\boldsymbol{h}(t)) \geq K_2 ||\nabla_{\boldsymbol{h}_i} E_\theta(\boldsymbol{h}(t))||^2$ then $||s_i(t)|| \geq K_2 ||\nabla_{\boldsymbol{h}_i} E_\theta(\boldsymbol{h}(t))||$.*

*Proof.* Noting that $||s_i(t)|| = ||\bar{s}_i(t)||$, we have

$$||s_i(t)|| \cdot ||\nabla_{\boldsymbol{h}_i} E_\theta(\boldsymbol{h}(t))|| \geq \bar{s}_i(t)^\top \nabla_{\boldsymbol{h}_i} E_\theta(\boldsymbol{h}(t)) \geq K_2 ||\nabla_{\boldsymbol{h}_i} E_\theta(\boldsymbol{h}(t))||^2. \tag{62}$$

Dividing both the left and right sides by $||\nabla_{\boldsymbol{h}_i} E_\theta(\boldsymbol{h}(t))||$ gives the desired result. $\square$

Having satisfied the assumptions, we can now apply the result that guarantees convergence.

**Proposition D.1 (Bertsekas and Tsitsiklis (1989), Proposition 5.1).** *Under Assumptions 2.1, D.1, and D.2, there exists some $\alpha_0 > 0$ (depending on $n$, $B$, $K_1$, and $K_3$) such that if $0 < \alpha < \alpha_0$ then $\lim_{t \to \infty} \nabla E_\theta(\boldsymbol{h}(t)) = 0$.*

# E  ASYNCHRONOUS GNN IMPLEMENTATION

In our asynchronous inference experiments, we simulate partially asynchronous execution (see Algorithm 1). We fix the maximum staleness bound to $B = 5$. Node updates are staggered across time and messages sent between nodes can incur delays. In particular, when a node updates, its next update time is selected randomly between time $t + 1$ and $t + S$, where $S := 5$ is the "stagger" time, and one of the last $D := 2$ values of its neighbors is chosen for performing the node update. This satisfies assumptions 1 and 2 of partial asynchronism; nodes update at least every $S$ time units, and messages a node's view of its neighbors is stale due to message delay by at most $D$ time steps.

# F  EXPERIMENT DETAILS (SYNTHETIC EXPERIMENTS)

## F.1  ARCHITECTURE DETAILS

For all implicitly-defined GNN architectures, we use the same node embedding size with $\boldsymbol{h}_i \in \mathbb{R}^2$. The architectures are chosen such that the number of parameters is approximately equal between models (with the constraint of using the same node embedding size). All architectures employ an output function $o_\phi$ which is parameterized as an MLP with layers $(4, 4, 1)$ ($(4, 4, 2)$ for the coordinates experiment, as node predictions are positions in $\mathbb{R}^2$).

**Energy GNN**  For energy GNN, we use a PICNN with layer sizes $(4, 4, 2)$ for the message function $m$ in Equation (17) and a PICNN with layer sizes $(4, 4, 1)$ for the update function $u$ in Equation (16). The aggregation in Equation (17) uses the entries of the unnormalized adjacency matrix $\boldsymbol{A}$ with no self-loops added. We add an independently parameterized self-loop to the message passing function $m$ (instead of using the same parameters as for neighbors). We set $\beta = 0.04$.

**Energy GNN + Attention**  For energy GNN with the attention mechanism, the attention weights $\alpha_{ij}$ are computed as in Equation (19); however, to maintain convexity with respect to the embeddings, the node and edge features (if present) are used in rather than using the embeddings $\boldsymbol{h}$. We use 2 attention heads, where the attention head outputs are concatenated to form the message $\boldsymbol{m}_i$ in Equation (17).

**GSDGNN**  For GSDGNN (where Equation (8) describes the optimization objective for obtaining embeddings), we parameterize $g_\theta$ as an MLP with layer sizes $(16, 16, 16, 2)$. We use the symmetric renormalized Laplacian matrix $\tilde{\boldsymbol{L}} = \boldsymbol{I} - \tilde{\boldsymbol{A}} = \boldsymbol{I} - (\boldsymbol{D} + \boldsymbol{I})^{-\frac{1}{2}}(\boldsymbol{A} + \boldsymbol{I})(\boldsymbol{D} + \boldsymbol{I})^{-\frac{1}{2}}$ for the Laplacian regularization term, and set $\gamma = 1.0, \beta = 5.0$. (Note that gradient-based optimization of this objective function has a direct correspondence to the embedding update function of APPNP Gasteiger et al. (2019).)

**IGNN**  For IGNN (described in Equation (22)), we parameterize $g_\theta : \mathbb{R}^{n \times p} \to \mathbb{R}^{n \times k}$ as an MLP with layer sizes $(16, 16, 16, 2)$.

**GCN and GAT**  For GCN, we use 5 layers of message passing with layer sizes $(10, 10, 10, 10, 10)$. For GAT, we use 5 layers of message passing with layer sizes $(3, 3, 3, 3, 3)$, and concatenate the output of 3 attention heads at each layer.

## F.2  TRAINING DETAILS

For binary classification experiments, we use binary cross entropy loss for training. For regression experiments, we use mean squared error. We use the Adam optimizer with weight decay, where we set the optimizer parameters as $\alpha = 0.001, \beta_1 = 0.9, \beta_2 = 0.999$. We set the learning rate to $0.002$, and use exponential decay with rate $0.98$ ever $200$ epochs. In the forward pass for IGNN, we iterate on the node update equation until convergence of node embeddings, with a convergence tolerance of $10^{-5}$. The maximum number of iterations is set to 500. In the forward pass for the optimization-based GNNs, we use L-BFGS to minimize $E_\theta$ w.r.t node embeddings, with a convergence tolerance of $10^{-5}$. The maximum number of iterations is set to 50. We train for a maximum of 5000 epochs. These experiments were performed on a single NVIDIA RTX 2080 Ti.

---

**Algorithm 1** Simulated asynchronous GNN inference

---

Initialize each node in $\mathcal{G}$ with all GNN parameters, its own node features $\boldsymbol{X}_i$, and for each of its neighbors $j$, the node and edge features $\boldsymbol{X}_j$, $\boldsymbol{E}_{ij}$, and weights $\boldsymbol{A}_{i,j}$. Let $L$ be the number of layers in the GNN, equal to $\infty$ for implicitly-defined GNNs. Let $T$ be the total number of simulated node updates. Let $n$ be the number of nodes in the graph. Let $S$ be the maximum number of time units between updates for any node, and $D$ be the maximum delay incurred by messages from neighbors.

**procedure** UPDATENODEOPT($\boldsymbol{H}$, $\boldsymbol{X}$, $\boldsymbol{E}$, $\boldsymbol{A}$, $\theta$, $t$, $i$)
    ▷ *Sum of nodes current view of neighbors' gradients* ◁
    $\boldsymbol{g}_i \leftarrow \sum_{j \in \mathcal{N}(i) \cup i} \boldsymbol{g}_{ji}(\tau_j^i(t))$
    ▷ *Node latent update is a single gradient step* ◁
    $\boldsymbol{h}_i \leftarrow \boldsymbol{h}_i - \alpha \boldsymbol{g}_i$
    ▷ *Node gradient is updated with new latent value* ◁
    $\boldsymbol{g}_{ij} \leftarrow \frac{\partial e_\theta^i}{\partial \boldsymbol{h}_j}(\boldsymbol{h}_j(\tau_j^i(t))), \qquad \forall j \in \mathcal{N} \cup i$

**procedure** UPDATENODEFINITE($\boldsymbol{H}$, $\boldsymbol{X}$, $\boldsymbol{E}$, $\boldsymbol{A}$, $\theta$, $t$, $i$)
    ▷ *Update node message and latent* ◁
    $\boldsymbol{m}_i \leftarrow \bigoplus_{j \in \mathcal{N}(i)} m^{t_i}\left(\boldsymbol{h}_i, \boldsymbol{h}_j(\tau_j^i(t)), \boldsymbol{x}_i, \boldsymbol{x}_j, \boldsymbol{e}_{ij}; \theta_m^{t_i}\right)$
    $\boldsymbol{h}_i \leftarrow u^{t_i}\left(\boldsymbol{m}_i, \boldsymbol{h}_i, \boldsymbol{x}_i; \theta_u^{t_i}\right)$

**procedure** SIMULATEASYNC($\boldsymbol{X}$, $\boldsymbol{E}$, $\boldsymbol{A}$, $L$, $T$, $n$, $S$, $D$)
    update_time $\leftarrow$ [ ]
    **for** $i = 1, \ldots, n$ **do**
        ▷ *Initialize current iteration count.* ◁
        $t_i \leftarrow 0$
        ▷ *Randomly select first update time in $(1, S)$* ◁
        update_time$_i \sim$ Uniform$(1, S)$
        update_time$[i] \leftarrow$ update_time$_i$
        **for** $j \in \mathcal{N}(i)$ **do**
            ▷ *Initialize staleness view of each neighbor* ◁
            $\tau_j^i \leftarrow 0$

    **for** $t = 0, \ldots, T$ **do**
        update_nodes $\leftarrow \{i = 1, \ldots, n \mid$ update_time$[i] == t\}$
        **for** $i \in$ update_nodes **do**
            **if** $t_i < L$ **then**
                **for** neighbors $j$ of node $i$ **do**
                    ▷ *Sample an updated stale view for node $j$* ◁
                    $(\tau_j^i)' \sim$ Uniform$(0, \min(t - \tau_j^i, D))$
                    $\tau_j^i \leftarrow t - (\tau_j^i)'$
                ▷ *Update the current node (finite GNN shown)* ◁
                UpdateNodeFinite($\boldsymbol{H}$, $\boldsymbol{X}$, $\boldsymbol{E}$, $\boldsymbol{A}$, $\theta$, $t_i$, $i$)

                ▷ *Randomly select next update time in $(0, S)$* ◁
                update_time$_i \sim$ Uniform$(1, S)$
                update_time$[i] \leftarrow t +$ update_time$_i$
                ▷ *Increment current iteration* ◁
                $t_i \leftarrow t_i + 1$
            **else**
                ▷ *Use readout to compute output* ◁
                $\hat{y}_i = o_\phi(h_i^L)$

## G  EXPERIMENT DETAILS (BENCHMARK DATASETS)

### G.1  DATASET DETAILS

The benchmark datasets we report performance for are MUTAG, PROTEINS, Peptides-func, and Peptides-struct, where the prediction task is graph classification/regression, and PPI, where the prediction task is node classification.

**MUTAG**   MUTAG is a dataset consisting of 188 graphs, each of which corresponds to a nitroaromatic compound (Srinivasan et al., 1996). The goal is to predict the mutagenicity of each compound on *Salmonella typhimurium*. Nodes in the graphs correspond to atoms (and are associated with a one-hot encoded feature in $\mathbb{R}^7$ corresponding to the atom type), and edges correspond to bonds. The average number of nodes in a graph is 17.93, and the average number of edges is 19.79.

**PROTEINS**   The PROTEINS dataset Borgwardt et al. (2005) consists of 1113 graphs, each of which corresponds to a protein. The task is predicting whether or not the protein is an enzyme. Nodes in the graph correspond to amino acids in the protein (and are associated with node features in $\mathbb{R}^3$ representing amino acid properties). Edges connect amino acids that are less than some threshold distance from one another in the protein. The average number of nodes is 39.06, and the average number of edges is 72.82.

**Peptides-func & Peptides-struct**   The peptides-func and peptides-struct datasets Dwivedi et al. (2022) consist of the same 15535 graphs, with different prediction targets. Each graph corresponds to a peptide; nodes correspond to heavy atoms (and are associated with 9 categorical node features representing atom properties) and edges correspond to bonds (and are associated with 3 categorical edge features representing bond properties). For peptides-func, the prediction task is multi-label graph classification (10 classes) of the peptide function. For peptides-struct, the prediction task is graph regression of various peptide properties (11 regression targets). The average number of nodes in a graph is 150.94, and average number of edges is 307.30. We use a train/valid/test split consistent with Dwivedi et al. (2022).

**PPI**   The PPI dataset (Hamilton, 2020) consists of 24 graphs, each of which corresponds to a protein-protein interaction network found in different areas of the body. Each node in the graph corresponds to a protein, with edges connecting proteins that interact with one another. Nodes are associated with features in $\mathbb{R}^{50}$, representing some properties of the protein. Each protein has 121 binary prediction targets, each of which corresponds to some ontological property that the protein may or may not have. We use a 20/2/2 train/valid/test split consistent with Hamilton et al. (2017).

### G.2  RESULTS

For all experiments with benchmark datasets, we use the same training procedure and architectures as described in Appendix F. For node classification tasks, the final layer of output function $o_\phi$ is modified to use layer sizes $(4, 4, (\text{num\_classes}))$ where num_classes is the number of distinct class labels. For graph classification/regression tasks, we obtain graph-level predictions by passing the mean of the node-level predictions through a graph readout function parameterized by an MLP with layers $(4, 4, (\text{num\_classes}))$.

For PROTEINS and MUTAG, we perform 10-fold cross validation and report average classification accuracy and standard deviations in Table 3. For PPI, we use a 20/2/2 train/valid/test split consistent with Hamilton et al. (2017), and report average micro-f1 scores in Table 4. For Peptides-func and Peptides-struct, we use a train/valid/test split consistent with Dwivedi et al. (2022) and report average precision and mean average error, respectively, in Table 5. Non-asterisked values correspond to our experimental setup, where the number of parameters (and embedding dimension) is equal across architectures. When parameter numbers are equal, the energy GNN architecture achieves the best performance on MUTAG, PPI, and Peptides-func, and achieves competitive performance on PROTEINS and Peptides-struct. We note that these reported results are under synchronous evaluation (similar to the synthetic experiments, Table 6 demonstrates that under asynchronous execution, we observe a decrease in performance for GCN and GAT, the representative explicitly-defined GNNs).

In Tables 3 and 4, we also include performance reported by other works (marked by an asterisk), which correspond to architectures using a higher number of parameters and larger embedding dimensions. Where layer specifications are not included (for asterisked values), we were unable to determine them from the cited

| | DATASET | |
|---|---|---|
| **MODEL** | **MUTAG** | **PROTEINS** |
| Energy GNN (edge-wise) | **87.6 ± 3.4** | 72.5 ± 0.3 |
| Energy GNN + attention | 79.5 ± 1.8 | 72.5 ± 0.5 |
| IGNN (1 layer) | 73.0 ± 0.6 | 71.8 ± 1.3 |
| GSD GNN | 78.4 ± 2.2 | 72.8 ± 0.8 |
| GCN (5 layer) | 78.0 ± 1.4 | **73.7 ± 0.5** |
| GAT (5 layer) | 76.1 ± 1.5 | 71.7 ± 3.2 |
| GCN* (Xu et al., 2019) (5 layer, embedding dimension=64) | 85.6 ± 5.8 | 76.0 ± 3.2 |
| IGNN* (Gu et al., 2020) (3 layer, embedding dimension=32) | 89.3 ± 6.7 | 77.7 ± 3.4 |
| GIN* (Xu et al., 2019) (5 layer, embedding dimension=64) | 89.4 ± 5.6 | 76.2 ± 1.9 |

Table 3: Graph classification accuracy (%). Results are averaged (and standard deviations are computed) using 10 fold cross validation with 5 random parameter seeds. Asterisked values are obtained from the work cited. Non-asterisked values correspond to architectures with the same number of parameters and 2-dimensional embeddings.

| **MODEL** | **micro f1** |
|---|---|
| Energy GNN (edge-wise) | **76.2** |
| Energy GNN + attention | 76.0 |
| IGNN (1 layer) | 75.5 |
| GSD GNN | 76.0 |
| GAT (5 layer) | 74.3 |
| GCN (5 layer) | **76.2** |
| MLP* (Gu et al., 2020) | 46.2 |
| GCN* (Gu et al., 2020) | 59.2 |
| GraphSAGE* (Veličković et al., 2018) (3 layer, embedding dimensions=[512, 512, 726]) | 76.8 |
| GAT* (Veličković et al., 2018) (3 layer, embedding dimension=1024) | 97.3 |
| IGNN* (Gu et al., 2020) (5 layer, embedding dimensions=[1024, 512, 512, 256, 121]) | 97.6 |

Table 4: Mean micro-F1 score for node classification on PPI dataset (%). Asterisked values are obtained from the work cited. Non-asterisked values correspond to architectures with the same number of parameters and 2-dimensional embeddings.

paper. Experiments with larger energy GNN architectures, including multi-layer energy GNN architectures, is left to future work.

| | DATASET | |
|---|---|---|
| **MODEL** | **Peptides-func** (AP) | **Peptides-struct** (MAE) |
| Energy GNN (edge-wise) | 0.348 | 0.367 |
| Energy GNN + attention | **0.381** | 0.402 |
| IGNN (1 layer) | 0.211 | 0.426 |
| GSD GNN | 0.343 | 0.375 |
| GCN (5 layer) | 0.362 | 0.366 |
| GAT (5 layer) | 0.355 | **0.335** |

Table 5: Performance on Peptides-func (average precision) and Peptides-struct (mean average error) datasets from the LRGB benchmarks. Values correspond to architectures with the same number of parameters and 2-dimensional embeddings.

| | DATASET | | | | |
|---|---|---|---|---|---|
| **MODEL** | **MUTAG** (%) | **PROTEINS** (%) | **PPI** (micro f1) | **Peptides-func** (AP) | **Peptides-struct** (MAE) |
| GCN (5 layer) | $16.0 \pm 10.2$ | $24.0 \pm 4.9$ | $25.1 \pm 9.5$ | $0.194 \pm 0.085$ | $0.734 \pm 0.041$ |
| GAT (5 layer) | $28.0 \pm 11.7$ | $40.0 \pm 5.8$ | $29.4 \pm 12.3$ | $0.494 \pm 0.031$ | $0.335 \pm 0.115$ |

Table 6: *Decrease* in task performance observed from switching from synchronous to asynchronous inference on sub-sample of test data (10 samples) using one trained model instance. Mean and standard deviation are across 5 asynchronous runs. The poor performance of GCN and GAT are consistent with the expected unreliability of explicitly-defined GNNs with asynchronous inference. Decreases in task performance for all implicitly-defined GNNs (IGNN, GSDGNN, and energy GNN variants) is less than $0.1\%$ of synchronous performance (i.e. result from numerical error); these values are omitted from the table for concision.

