# OpenReview forum: "Graph Neural Networks Gone Hogwild"
_ICLR.cc/2025/Conference — ICLR 2025 Poster_

### Official Review · Reviewer_M6Fj · 2024-10-31

**Soundness:** 4
**Presentation:** 3
**Contribution:** 3
**Rating:** 6
**Confidence:** 3

**Summary:**

In this paper, the authors tackle the issue of asynchrony in graph neural networks (GNNs). Traditional GNNs, particularly multi-layer variants, assume synchronized node updates, which is often unrealistic in real-world distributed systems like robotic swarms or sensor networks.

The authors identify a class of GNNs called "implicitly-defined" GNNs that are robust to asynchronous or "hogwild" inference. They propose a novel implicitly-defined GNN architecture called "energy GNN", which leverages input-convex neural networks to parameterize a convex energy function. The results from experiments on synthetic multi-agent tasks and benchmark graph datasets demonstrate the superior performance of energy GNNs under asynchrony and their competitiveness even in synchronous settings.

**Strengths:**

1) Adoption of GNNs in real-world distributed systems.
2) Theoretical guarantees for the robustness of implicitly-defined GNNs to asynchrony, drawing on concepts from distributed optimization.
3) The proposed energy GNN offers a flexible and expressive way to define convex energy functions, potentially leading to more powerful GNN models.
4) Experiments on both synthetic (chains, counting, sums, coordinates) and benchmark datasets (MUTAG, PROTEINS, PPI) demonstrate the effectiveness of the proposed approach.

**Weaknesses:**

1) Even though the paper proposes mitigation strategies for trraining implicitly-defined GNNs. They are computationally expensive due to the iterative nature of the forward pass. This would be a challenge in practical scenarios.
2) The experiments primarily focus on single-layer energy GNNs. The performance of multi-layer variants and their scalability to larger graphs are unclear.
3) The convergence of implicitly-defined GNNs can be sensitive to the choice of hyperparameters like step size and convergence tolerance. Have the authors investigated the robustness perspective?
4) While the paper motivates the problem with real-world scenarios(in abstract) the experiments are primarily on synthetic datasets. Unsure how well these would be effective in real world scenarios due to few of the weaknesses pointed above

**Questions:**

1) How does the choice of the staleness bound B and stagger time S affect the performance and convergence of energy GNNs in practice? Is there a principled way to select these parameters?
2) The authors mention the potential for numerical instability due to ill-conditioned Hessians or Jacobians. Have they explored techniques like preconditioning to address this issue?
3) How does the performance of energy GNNs scale with the size of the graph and the number of nodes? Would these be applicable to very large graphs? (I see a comment on this in conclusion section but would be interesting to know the authors intuition about this)

---

> ### Author Response · Authors · 2024-11-23
>
> Thanks for your review of our work. We address your questions below:
>
> 1. In our simulations of asynchronous inference, the staleness bound accounts for simulated delay or loss of messages, and stagger time of updates accounts for the level of asynchrony in node updates. These are parameters which are inherent to the distributed system under operation and are not controllable (assuming a fixed hardware setup). These values could be inferred by collecting telemetry from the system of interest. We essentially arbitrarily choose values for these bounds in our asynchronous simulation, without reference to a particular physical system. In theory and in practice, the higher the values of these bounds is, the slower the system will be to converge. However, they do not affect the performance of the system, in the sense that eventually the embeddings will converge to the same values regardless of bounds.
>
> 2. We did consider preconditioning; this could help with the linear system in the backward pass, but would be expensive since the preconditioner would need to be recomputed for every backward pass. This is because the Jacobian/Hessian depends on the model parameters and embeddings, which change over time. Furthermore, for optimization-based GNNs, poor conditioning of the Hessian causes difficulty not only in the linear system solve in the backward pass, but also in minimization of E_\theta in the forward pass (this is based on the simple fact that gradient-based optimization is more difficult for functions with poorly conditioned Hessians). Since the forward pass is a convex optimization problem, and cannot be framed as a linear system, preconditioning is unfortunately not applicable. We are also exploring other implicitly-defined GNN architectures in which the condition number of the Hessians/Jacobians could be more tightly controlled.
>
> 3. All implicitly-defined GNNs scale more poorly to large graphs than explicitly-defined GNNs due to the fact that the backward pass requires solving a linear system (which scales linearly with the number of nodes), and due to the iterative nature of both the forward and backward pass. That being said, there is nothing preventing implicitly-defined GNNs from being scaled to handle large graphs through distributed training approaches. We don’t implement distributed training in our work since the graphs in the experiments are relatively small compared to benchmark GNN datasets which require distributed training (but they are typical sizes for multi-robot systems - for instance, in the referenced papers where GNNs have been applied to decentralized multi-agent systems, the number of agents range from tens to at most hundreds).

---

> > ### Comment · Reviewer_M6Fj · 2024-11-25
> >
> > Thank you for providing detailed answers to my questions. This gives me a better understanding of the work and I believe this would be a good paper. I have also read the other reviews and there are still few concerns that needs to be addressed. I will maintain my score for now!

---

> > > ### Author Response · Authors · 2024-11-25
> > >
> > > Thanks for your response - could you clarify what additional concerns of yours need to be addressed?

---

> > > > ### Comment · Reviewer_M6Fj · 2024-11-27
> > > >
> > > > Appreciate the follow-up. I am more inclined to the "realistic evaluations" comment from other reviewer. Would definitely be worth seeing that in the paper. If the authors have further thoughts on that please mention here.

---

### Official Review · Reviewer_9p4E · 2024-11-02

**Soundness:** 3
**Presentation:** 2
**Contribution:** 2
**Rating:** 5
**Confidence:** 4

**Summary:**

This paper introduces Graph Neural Networks (GNNs) for distributes multi-agent systems with asynchronous communication. Traditional GNNs struggle with asynchronous execution and unreliable communication. This results in unreliable predictions. To address this, the authors focus on implicitly-defined GNNs, a class of models that can handle partial asynchrony. They introduce a model called energy GNNs, which outperform existing implicitly-defined GNNs on synthetic multi-agent tasks.

The experimental results highlight potential applications for GNNs in control tasks and real-time inference on dynamic graphs. This is an important property for multi-robot systems. The paper also notes the training limitations for implicitly-defined GNNs, particularly in achieving convergence, as it requires complex and often unpredictable computations. Strategies like warm-start initialization and implicit differentiation could help with these challenges, but challenges remain in achieving stability.

**Strengths:**

By focusing on implicitly-defined GNNs, the work addresses a major limitation of conventional GNNs in handling asynchronous and unreliable communication, making it highly relevant for real-world multi-agent systems.

Experimental results show that energy GNNs outperform other implicitly-defined GNNs on synthetic tasks, providing empirical validation for the architecture's effectiveness in multi-agent tasks (although most experiments are toy-ish.)

**Weaknesses:**

The experiments are conducted on toy examples.

For the experiments other than the "terrain" examples there are great solutions that do not require machine learning. The results on the benchmark datasets in the supplementary show small improvements as compared to the more toy examples in the main manuscript.

**Questions:**

Please consider changing the term energy GNN. My first reaction was that this solution was energy-efficient which is a very big concern in AI at the moment. However this work is not about energy efficient AI.

The experiments are on toy examples that do not need machine learning. Please expand the experimental range to other problems and to larger problem scale.

Can you be clearer about what you summarize from existing work and what are the new contributions in sections 2, 3, 4

The description of energy GNN is very sparse, please explain the computation of the neuron and node, architecture, and the method for training in the main manuscript. Please address summarize the properties of this model in Section 5.

Can you say anything about the performance and energy consumption of these models?

---

> ### Author Response · Authors · 2024-11-23
>
> Thanks for taking the time to review our work. We address your questions below:
>
> 1. We choose the name energy GNN due to the correspondence to energy-based models, where predictions result from energy minimization (i.e. [LeCun et al 2006]. A Tutorial on Energy-Based Learning).
> 2. The graphs in our synthetic experiments are relatively small compared to benchmark GNN datasets, but they are typical sizes for multi-robot systems - for instance, in the referenced papers where GNNs have been applied to decentralized multi-agent systems, the number of agents range from tens to at most hundreds. While some of the tasks do not require machine learning, they still have value and serve as a sanity check since it is known that they are solvable in decentralized and asynchronous settings. Regarding expanding to other problems - there are no standard benchmarks for applying GNNs to graphs describing multi-agent systems, which is the motivation behind the work. Standard GNN benchmark datasets do not correspond to the kind of tasks multi-agent systems are expected to perform. We believe the suite of synthetic experiments we propose make sense in the context of our motivation.
> 3. Section 2 is entirely background material. In section 3, we define implicitly-defined GNNs and contextualize existing architectures that fall under this definition. Section 4 is entirely novel, where we derive GNN updates under partial asynchrony and summarize results on convergence (for which a detailed proof is included in the appendix).
> 4. We believe we have summarized the main properties of the proposed energy GNN architecture in the final paragraph of section 5. The main details left to the appendix are the architecture of the PICNNs which make up the message function $m()$ and the update function $u()$; these are in the appendix since we simply restate information from the original work on Input Convex Neural Networks. Training details are provided in the experimental setup in section 6, as well as in the appendix.
> 5. We can qualitatively say that training implicitly-defined GNNs is more time and resource intensive than training explicitly-defined GNNs due to the iterative nature of the forward and backward pass. We aren’t sure what kind of quantitative measures you’re asking for regarding energy consumption, but generally energy consumption is non-trivial to measure and requires, e.g. hardware probes or simulations of execution on particular hardware.

---

> > ### Comment · Reviewer_9p4E · 2024-11-23
> >
> > Thank you for the feedback. While I understand the alignment between the graph sizes in your synthetic experiments and typical multi-robot systems, it remains a concern that these smaller scales may limit the broader applicability of your findings to larger, more complex scenarios. It would be helpful to clarify how your work might extend beyond these constraints and whether scalability has been considered in the design of your experiments.
> >
> > The lack of standard benchmarks for applying GNNs to multi-agent systems is a valid point, but it also raises the question of whether the synthetic tasks proposed fully capture the diversity and complexity of real-world multi-agent challenges. While these tasks may serve as sanity checks, their utility as meaningful benchmarks could benefit from further justification, especially given the broader aims of your work.
> >
> > How do you see these experiments evolving into more representative or widely accepted benchmarks in the future? Additionally, are there plans to validate your approach on larger or more realistic systems to strengthen its generalizability?

---

> > > ### Author Response · Authors · 2024-11-25
> > >
> > > Thanks for your response to our rebuttal. We discuss your outstanding concerns below:
> > >
> > > We completely agree that the synthetic experiments do not fully capture the diversity and complexity of real-world multi-agent challenges. Doing this involves a full hardware in the loop experiment in the field, and/or a full-featured simulator. Most published works (eg. PettingZoo) use simulated environments and case studies; they operate in simplified or abstracted variations of complete applications. This is often a feature of these works, and not necessarily a bug. Serious and complete treatments of most applications are often in conflict with clear and precise contributions to a particular modular component of the application. To be concrete and specific with respect to our work, we imagine an asynchronous-capable GNN (e.g., an Energy GNN) to serve as a method for generating state representations in a full-fledged adaptive multi-agent control application. Our experiments are essentially computational kernels drawn from these applications; small components of a complete system which is capable of long-range communication (sums, counting), localization (coordinates), and estimation (MNIST terrain). There does not exist a standardized or accepted collection of these kernels/benchmarks, so there is some subjectivity involved in their selection, but we think that the case studies we present are relevant and informative inclusions.
> > >
> > > We agree that conducting more experiments is in some sense always desirable. But, we think that the experiments we’ve presented are a good starting point, and they provide evidence for important claims: 1) our empirical results confirm our theoretical findings about robustness to asynchronous execution for implicitly-defined GNNs, 2) implicitly-defined GNNs are capable of capturing long-range dependencies, 3) implicitly-defined GNNs are capable at tasks that existing, bespoke decentralized algorithms are designed for, 4) the energy GNN architecture we propose outperforms other existing implicitly-defined GNNs. These positive results strongly motivate further experiments and consideration of implicitly-defined GNNs in decentralized and asynchronous multi-agent settings.
> > >
> > > We expect that a more 'realistic' suite of benchmarks could be created by considering end-to-end learning in multi-agent control tasks, which we mention in the discussion section of our paper. In our synthetic experiments, we consider static prediction tasks (and we note again that there are no standard benchmarks for these tasks) rather than control tasks. Again, the reason we do not immediately jump to control tasks on dynamic graphs is because there are many moving parts in these kinds of tasks, and we believe a more modular approach which first considers static prediction is a better way of evaluating implicitly-defined GNNs in isolation.
> > >
> > > Finally, we want to nod to the unfortunate zero-sum regime of including additional (or more complex) experiments and achieving clarity and completeness in the presentation of our analytical results. To make precise and justify much of what we have to say about asynchrony in GNNs requires mathematical infrastructure and arguments. We aimed to find a balance between an understandable and correct presentation of those new ideas, and abstracted but informative case studies to empirically ground this analysis. In brief, we didn’t consider complete applications in scope for this single 10-page work, as we’d neither be able to give a considerate treatment of the application in full, nor a clear and thorough justification of our claims. We think that leaving this to future work does not detract from the main contributions of the paper: we shed light on problems in applying typical GNNs to decentralized multi-agent systems, and propose a theoretically grounded solution.

---

> > > > ### Comment · Reviewer_9p4E · 2024-11-27
> > > >
> > > > OK thank you for this additional input. I will increase the score to 6; I can see value in bringing these ideas to the community but for these ideas to be truly relevant you need more serious and realistic evaluations.

---

### Official Review · Reviewer_FLMf · 2024-11-04

**Soundness:** 3
**Presentation:** 4
**Contribution:** 2
**Rating:** 8
**Confidence:** 4

**Summary:**

The paper studies GNNs in the asynchronous “hogwild” setup – when node states are not updated simultaneously at the same time. Such setup is common in distributed environments, agentic systems, and temporal learning where enforcing synchronous node updates might be impossible or too expensive to maintain. Standard, explicitly-defined GNNs trained in the synchronous mode fail in the async mode, so the authors turn their attention to implicitly-defined GNNs (further categorized into fixed-point and optimization-based GNNs).

It is shown that implicit GNNs are robust to partial asynchronicity. Motivated by the theoretical findings, the authors propose EnergyGNN - an optimization-based implicit GNN where node states minimize a convex energy function. The space of possible energy functions is rather wide and allows for node-wise, edge-wise, and attention-based parameterizations. Experimenting on a range of synthetic tasks, the proposed EnergyGNN outperforms other implicit GNNs in the synchronous regime, is robust to the delayed node update setup, and is on par with synchronous baselines on MUTAG and Proteins datasets.

**Strengths:**

**S1**. A theoretical study on the async inference with GNNs is timely and important - many real-world tasks are of that nature, so having a principled, robust approach for such problems (instead of tinkering standard sync GNN architectures) might be of interest to the graph learning community.

**S2**. The paper is well-written - complex concepts are properly introduced and explained (which is often a challenge in the literature on implicit GNNs), the story and motivation are easy to follow.

**Weaknesses:**

The main problem of the work is in the experiments - it is hard to judge the claimed effectiveness of the proposed EnergyGNN using only synthetic experiments and basic GCN / GAT as baselines.

**W1**. In the proposed suite of tasks, all implicit GNNs are robust in the async setup (the main goal of the work), and the main difference lies in the performance in the sync setup. Is there a different way to evaluate the differences among implicit GNNs other than on sync tasks? EnergyGNN is better than other implicit models but comparing against vanilla GCN and GAT on benchmarks defined to be of a long-range nature (where vanilla GNNs are bound to fail) is questionable. There is a variety of explicitly-defined GNNs that might be stronger baselines in such setups like Half-Hop [1], DRew [2] and other graph rewiring methods, as well as various graph transformers from the dedicated Long-Range Graph Benchmark [3].

**W2**. The proposed synthetic benchmarks are rather small and might not correlate with the performance on real datasets where async inference is important (or MUTAG and Proteins with sync inference); it seems to be a stretch to attribute chains, node counting, and node sums to “agentic” tasks. Instead, experiments on more real benchmarks might be more informative and evidential:
* Since some tasks focus on the long-range dependencies, LRGB [3] is a suitable choice;
* As async inference implies nodes appearing and disappearing at some moments of time, temporal graph benchmarks with many snapshots [4] might be a great choice.

Generally, I am willing to increase the score if the authors add more modern baselines and/or real-world datasets.

[1] Azabou et al. Half-Hop: A graph upsampling approach for slowing down message passing. ICML 2023.
[2] Gutteridge et al. DRew: Dynamically Rewired Message Passing with Delay. ICML 2023.
[3] Dwivedi et al. Long Range Graph Benchmark. NeurIPS 2022.
[4] Huang et al. Temporal Graph Benchmark for Machine Learning on Temporal Graphs. NeurIPS 2023.

**Questions:**

Stemming from the weaknesses:

**Q1 (W1)**. Is there a different way to evaluate the differences among implicit GNNs other than on sync tasks?
**Q2 (W1)**. How strong is EnergyGNN compared to more long-range optimized GNNs like Half-Hop, DRew, and graph transformers?
**Q3 (W2)**. Does the synthetic EnergyGNN performance correlate with the tasks from more real-world benchmarks like LRGB and TGB?

---

> ### Author Response · Authors · 2024-11-23
>
> Thanks very much for the review, and particularly for providing some criteria for updating your score.
>
> 1. Regarding additional ways to compare implicitly-defined GNNs (besides their synchronous performance): implicitly-defined GNNs could be compared on the basis of how quickly node embeddings converge. However, the speed of convergence depends on where node embeddings are initialized. A fair comparison could be made by evaluating implicitly-defined GNNs in the context of real-time inference on dynamic graphs and observing the number of iterations required for convergence when the graph changes from one state to the next. For instance, for the MNIST terrain task, nodes could be allowed to ‘boot up’ and iterate on their embeddings starting from random node embedding initialization. Following convergence, the graph could be modified by allowing the nodes to take a step on the image “terrain”. The number of iterations required for embeddings to converge following this step would then be a measure of how quickly nodes are able to respond to changes in their observations. We intend to perform this kind of evaluation in future work, where dynamic graphs are considered.
> 2. (&3) The main motivation of the paper is applying GNNs in multi-agent systems, where decentralized and asynchronous execution is desirable. We don’t perform comparisons against explicitly-defined SOTA GNNs because they cannot run asynchronously, making them unsuitable for the applications we’re interested in. We perform experiments with GCN and GAT for illustrative purposes, using them as representatives of explicitly-defined GNNs to demonstrate that under asynchrony the predictions computed by this class of GNNs are unreliable. Under asynchronous conditions, if reliable and consistent predictions are required (which is the setting that we consider), one is restricted to using implicitly-defined GNNs, so comparison between these models is the focus of our experiments.
> Due to time constraints, we chose 2 tasks from the LRGB benchmarks that you suggest; Peptides-func and Peptides-struct. Results for these tasks is summarized here:
>
> Table 1. Synchronous performance on Peptides-func and Peptides-struct test data.
> | Model    | Peptides-func (AP, higher is better) | Peptides-struct (MAE, lower is better)
> | -------- | ------- | ------- |
> | Energy GNN (edge-wise)  | 0.348  | 0.367
> | Energy GNN + attn | 0.381    | 0.402
> | IGNN  |  0.211    | 0.426
> | GSDGNN  | 0.343    |  0.375
> | GCN  | 0.362   | 0.366
> | GAT | 0.355   | 0.335
>
> Table 2. Decrease in performance on sample of 10 test graphs from Peptides-func and Peptides-struct datasets, mean and standard deviation computed over 5 random asynchronous runs.
> | Model    | Peptides-func | Peptides-struct
> | -------- | ------- | ------- |
> | GCN  | 0.194 $\pm$ 0.085   | 0.734 $\pm$ 0.041
> | GAT | 0.494 $\pm$ 0.031   | 0.335 $\pm$ 0.115
>
>
> We first note that we have shown again that explicitly-defined GNNs drop in performance when executed asynchronously. This again justifies our decision not to run against more SOTA explicitly-defined GNNs, given our interest in asynchronous inference; no matter how good SOTA GNNs perform synchronously, they simply are not suitable in asynchronous settings. We also note that performance is not competitive with reported results from the literature using SOTA explicitly-defined GNNs; this is to be expected, since the models we use are small and have on the order of ~1000 parameters, while SOTA GNNs use on the order of ~500K parameters. Our model is inspired by and designed for distributed, decentralized, and often extremely resource constrained applications (imagine microcontrollers or very small processors on robotic platforms in the field). Standard GNN datasets are not the intended use case for our model (in this work), so we did not perform experiments with large scale models. Unfortunately, scaling the size of our model (and the other implicitly-defined GNNs) for the proposed benchmarks is not feasible given the time constraints for the rebuttal; the time required to train a ~500K parameter explicitly-defined GNN on the suggested datasets is at least 60hr (taken from the paper introducing the LRGB dataset), and implicitly-defined GNNs would certainly be expected to require more time.
> All that said, we want to flag our shared interests and inclination toward (1) long range dependencies and in particular (2) (temporally) dynamic graphs. We too are excited and motivated by these areas, particularly the latter (the former is actually the motivation behind work introducing the IGNN architecture), and suspect that Energy GNNs could be a good fit. While these delineations are admittedly subjective, we believe consideration of dynamic graphs constitutes a different work in full, and could not be meaningfully or seriously treated as an additional component of our current work (not least in a single 10 page document). But, we do intend to pursue these avenues in the future.

---

> > ### Author Response · Authors · 2024-11-23
> >
> > As a side note about the size of the synthetic experiments, which you mention in the weaknesses: the graphs are relatively small compared to many benchmark GNN datasets, but they are typical sizes for multi-robot systems - for instance, in the referenced papers where GNNs have been applied to decentralized multi-agent systems, the number of agents range from tens to at most hundreds.

---

> > > ### Comment · Reviewer_FLMf · 2024-11-23
> > >
> > > Thank you for the responses and new experiments on LRGB - most of my concerns have been addressed and I increased the score. I would encourage the authors to include those results into the main body of the paper and perhaps add initial experiments on dynamic graphs into the camera-ready version.

---

> > > > ### Author Response · Authors · 2024-11-27
> > > >
> > > > Thanks for your response, and for increasing your score. We've uploaded a revised pdf with the Peptides-func and Peptides-struct results included.

---

### Official Review · Reviewer_GQeF · 2024-11-04

**Soundness:** 3
**Presentation:** 2
**Contribution:** 3
**Rating:** 6
**Confidence:** 2

**Summary:**

This paper investigates the performance of Graph Neural Networks (GNNs) in partially asynchronous inference settings, where nodes update in a staggered or asynchronous manner. The authors categorize GNNs into two types: "explicitly-defined" and "implicitly-defined." They demonstrate that explicitly-defined GNNs are highly vulnerable to asynchronous updates. In contrast, implicitly-defined GNNs are shown to be robust. Additionally, the authors propose a novel explicitly-defined model, termed Energy GNN, which achieves notable improvements over existing methods on synthetic datasets.

**Strengths:**

- Although I am not deeply familiar with the literature on asynchronous inference in GNNs, the proposed Energy GNN model introduces a novel and meaningful contribution to this field.
- The paper is of high quality. The authors provide comprehensive mathematical proofs for the convergence of Energy GNNs. They also provide a detailed description of the experimental setup and results, which are well-organized and easy to follow.
- The paper is generally well-structured, with a clear definition of the problem space and a concise summary of related GNN methods. The paper clearly defines the problem of asynchronous inference in explicitly-defined GNNs.
- The proposed Energy GNN model demonstrates significant improvements on synthetic datasets, and competitive performance on real-world datasets.

**Weaknesses:**

The paper lacks a clear and intuitive explanation of implicitly-defined GNNs, which is essential for understanding their robustness to asynchronous updates. While the authors offer detailed explanations for explicitly-defined GNNs, which are more straightforward, they do not provide the same depth of insight into implicitly-defined GNNs. This makes it difficult for readers unfamiliar with the topic to understand how implicitly-defined GNNs work and why they are resilient to asynchronous inference.

Additionally, although the proposed Energy GNN shows strong results on synthetic datasets, its performance on real-world datasets is rather competitive. On the PPI dataset, in particular, its performance is comparable to that of explicitly-defined GNNs, which were expected to fail under asynchrony. This discrepancy between synthetic and real dataset performance is not explained. A broader evaluation across various real-world datasets would increase the credibility of Energy GNNs as a robust solution.

**Questions:**

- Could you provide a more intuitive explanation of implicitly defined GNNs, specifically highlighting the mechanisms contributing to their robustness under asynchronous updates?
- Do you have any insight into why the performance on real-world datasets is not as great as on synthetic data?
- Can you offer an explanation or hypothesis as to why explicitly defined GNNs did not perform as poorly as expected on the PPI dataset under asynchronous conditions?

Line 369. There appears to be a typo in the index notation within the description of targets for the Sums dataset.

---

> ### Author Response · Authors · 2024-11-23
>
> Thank you for your review of our work. We address each of your questions below:
>
> 1. Implicitly-defined GNNs fall into two categories: fixed-point GNNs and optimization-based GNNs. Fixed-point GNNs use iterated function application to produce a sequence $x_0, f(x_0), f(f(x_0)), \dots$ whose limit serves as the output of the layer (the node embeddings). This function depends on the data and the graph structure, and is constructed in such a way that the sequence of iterated applications converges (namely, by making $f$ a contraction map). Optimization-based GNNs define node embeddings as the minimizers of a convex function defined over the graph, and use a convex optimization procedure (e.g. gradient descent) to obtain embeddings. Robustness to asynchrony can be provably shown for contractive functions, and for gradient-based optimization of strongly convex functions (with bounded Hessian norms) provided the step size is small enough (we provide a proof in the appendix).
> You can think of this (roughly) as a statement about stability over coordinate-wise update ordering. For instance, suppose we are tasked with minimizing a function in $\mathbb{R}^2$, but we are only allowed to update one coordinate at a time. In the optimization-based GNN context, think of a graph with two nodes, whose scalar embeddings we want to choose. If the function is convex, its derivatives are bounded, we take small enough steps, and we never update just one coordinate too many times in a row, we can still minimize the function (under many particular sequences of updates). These are the (informal renditions) of the kinds of technical conditions that, if fulfilled by an implicit GNN, guarantees robustness.
> 2. (&3) There may be a misunderstanding re: the benchmark (real-world) dataset performance. Importantly, in the benchmark dataset results we report, the associated model is **not** executed asynchronously. So, this is the "normal" operating regime of an explicitly-defined GNN, we wouldn't expect them to fail. Like for the synthetic experiments, if we had performed asynchronous simulations for the benchmark datasets, we would have seen that the explicitly-defined GNNs had unreliable performance while the implicitly-defined GNNs had performance consistent to synchronous execution. What we aimed to signal with those benchmark results is: even run synchronously on standard/typical GNN benchmarks, our model is reasonably competitive. We don't mean to over-emphasize those results, or claim anything like our model being state of the art on typical GNN benchmarks.

---

> > ### Comment · Reviewer_GQeF · 2024-11-27
> >
> > Thank you for your reply and the explanation.
> >
> > Since no revised pdf was uploaded, I'll maintain my rating.

---

> > > ### Author Response · Authors · 2024-11-27
> > >
> > > We've included a statement in the appendix section on benchmark datasets, explaining that results are shown under synchronous execution (i.e. Q2/Q3 are now addressed in the appendix). We believe that the response to Q1 in our previous response is simply a summary of information in sections 3 and 4 of the main paper. Could you clarify what additional revisions you'd like to see to the paper?

---

### Meta-Review · Area_Chair_m7pH · 2024-12-20

**Metareview:**

The paper addresses the issue of graph networks (GNNs) that this networks generate incorrect predictions if nodes update asynchronously. This is an issue in distributed, decentralized multi-agent systems, where it is not feasible to update the nodes at the same time. The authors propose implicitly-defined GNNs that are robust to asynchronous updates and can be used in distributed systems. The paper provides comprehensive mathematical proofs for the convergence of Energy GNNs and demonstrates significant, or competitive, improvements on synthetic datasets and real-world datasets. As a weakness, most experiments were provided on small synthetic datasets, though the target graphs in multi-robot systems are usually small as well. I recommend for Accept (poster).

**Additional Comments On Reviewer Discussion:**

The reviewers were concerned about the experiments on small, synthetic datasets, and the results might not correlate with the performance on real datasets where asynchronous inference is important. During the rebuttal, the authors provided the new results on  Peptides dataset and noted that the target applications  in multi-robot systems also tend to have small size. Authors also expanded the discussion about applications to real world dataset.

We suggest the authors to include clarifications to reviewer GQeF into the camera-ready version.

---

### Decision · Program_Chairs · 2025-01-22

Accept (Poster)